# Alcohol intake and pancreatic cancer risk: An analysis from 30 prospective studies across Asia, Australia, Europe, and North America

Sabine Naudin[1,2], Molin Wang[3,4,5], Niki Dimou[1], Elmira Ebrahimi[1], Jeanine Genkinger[6,7], Hans-Olov Adami[8,9], Demetrius Albanes[10], Ana Babic[11], Matt Barnett[12], David Bogumil[13], Hui Cai[14], Chu Chen[15], A. Heather Eliassen[3,16], Jo L. Freudenheim[17], Gretchen Gierach[10], Edward L. Giovannucci[3,16], Marc J. Gunter[1], Niclas Håkansson[18], Mayo Hirabayashi[19], Tao Hou[16], Brian Z. Huang[13], Wen-Yi Huang[10], Harindra Jayasekara[20,21,22], Michael E. Jones[23], Verena A. Katzke[24], Woon-Puay Koh[25,26], James V. Lacey[27], Ylva Trolle Lagerros[28,29], Susanna C. Larsson[18], Linda M. Liao[10], Kenneth Lo[30], Erikka Loftfield[10], Robert J. MacInnis[20,21], Satu Männistö[31], Marjorie L. McCullough[32], Anthony Miller[33], Roger L. Milne[20,21,34], Steven C. Moore[10], Lorelei A. Mucci[3,35], Marian L. Neuhouser[12], Alpa V. Patel[32], Elizabeth A. Platz[36,37], Anna Prizment[38], Kim Robien[39], Thomas E. Rohan[40], Carlotta Sacerdote[41], Sven Sandin[8,42,43], Norie Sawada[44], Minouk Schoemaker[45], Xiao-Ou Shu[14], Rashmi Sinha[10], Linda Snetselaar[46], Meir J. Stampfer[3,16,5], Rachael Stolzenberg-Solomon[10], Cynthia A. Thomson[47], Anne Tjønneland[48], Caroline Y. Um[32], Piet A. van den Brandt[49], Kala Visvanathan[36,37], Sophia S. Wang[27], Renwei Wang[50], Elisabete Weiderpass[1], Stephanie J. Weinstein[10], Emily White[51], Walter Willett[3,16], Alicja Woslk[18], Brian M. Wolpin[11,5], Shiaw-Shyuan S. Yaun[16], Chen Yuan[11], Jian-Min Yuan[50,52], Wei Zheng[14], Paul Brennan[1], Stephanie A. Smith-Warner[3,16‡], Pietro Ferrari[1‡*]

1 International Agency for Research on Cancer, World Health Organization, Lyon, France, 2 UPS, UVSQ, National Institute of Health and Medical Research, Gustave Roussy, Centre for research in epidemiology and population health, Villejuif, France, 3 Department of Epidemiology, Harvard TH Chan School of Public Health, Boston, Massachusetts, United States of America, 4 Department of Biostatistics, Harvard TH Chan School of Public Health, Boston, Massachusetts, United States of America, 5 Channing Division of Network Medicine, Department of Medicine, Brigham and Women's Hospital and Harvard Medical School, Boston, Massachusetts, United States of America, 6 Department of Epidemiology, Mailman School of Public Health, Columbia University Medical Center, New York, New York, United States of America, 7 Cancer Epidemiology Population Sciences Program, Herbert Irving Comprehensive Cancer Center, Columbia University Medical Center, New York, New York, United States of America, 8 Department of Medical Epidemiology and Biostatistics, Karolinska Institutet, Stockholm, Sweden, 9 Clinical Effectiveness Group, Institute of Health and Society, University of Oslo, Oslo, Norway, 10 Division of Cancer Epidemiology and Genetics, National Cancer Institute, Rockville, Maryland, United States of America, 11 Department of Medical Oncology, Dana-Farber Cancer Institute and Harvard Medical School, Boston, Massachusetts, United States of America, 12 Division of Public Health Sciences, Fred Hutchinson Cancer Center, Seattle, Washington, United States of America, 13 Department of Population and Public Health Sciences, Keck School of Medicine of USC, Los Angeles, California, United States of America, 14 Division of Epidemiology, Department of Medicine, Vanderbilt University School of Medicine, Nashville, Tennessee, United States of America, 15 Program in Epidemiology, Division of Public Health Sciences, Fred Hutchinson Cancer Center, Seattle, Washington, United States of America, 16 Department of Nutrition, Harvard TH Chan School of Public Health, Boston, Massachusetts, United States of America, 17 Department of Epidemiology and Environmental Health, School of Public Health and Health Professions, University at Buffalo, Buffalo, New York, United States of America, 18 Department of Environmental Medicine, Karolinska Institutet, Stockholm, Sweden, 19 Division of Prevention, National Cancer Center Institute for Cancer Control, Tokyo, Japan, 20 Cancer Epidemiology Division, Cancer Council Victoria, Melbourne, Victoria, Australia, 21 Centre for Epidemiology and Biostatistics, Melbourne School of Population and Global Health, The University of Melbourne, Melbourne, Victoria, Australia, 22 School of Public Health and Preventive Medicine, Faculty of Medicine, Nursing and Health Sciences, Monash University, Clayton, Victoria, Australia, 23 Division of Genetics and Epidemiology, The Institute of Cancer Research, London, United Kingdom, 24 Division of



**Data availability statement:** Consortium data can be made available upon request to the EPIC website (https://epic.iarc.who.int/contact-us/). The sharing of data will be governed by executed data use agreements between the home institution of each cohort studies and the Harvard T.H. Chan School of Public Health, as well as approval from lead investigators for each cohort and/or the cohort study's leadership. Statistical programs can be made available upon request to the corresponding author.

**Funding:** The centralization, checking, harmonization, and statistical analyses of the participant level data from the cohorts were supported by NIH AAA R01 grant (grants: R01AA024770, GR-IARC-2015-05-05-01 to PF and SASW). The funders had no role in study design, data collection and analysis, decision to publish, or preparation of the manuscript. Where authors are identified as personnel of the International Agency for Research on Cancer/World Health Organization, the authors alone are responsible for the views expressed in this article and they do not necessarily represent the decisions, policy or views of the International Agency for Research on Cancer/ World Health Organization. Alpha-Tocopherol Beta-Carotene Cancer Prevention Study (ATBC): The ATBC study is supported by the Intramural Research Program of the U.S. National Cancer Institute, National Institutes of Health, Department of Health and Human Services. Breast Cancer Detection Demonstration Project Follow-Up Study (BCDDP): The BCDDP Study is supported by the Intramural Research Program of the Division of Cancer Epidemiology and Genetics, U.S. National Cancer Institute (NCI), National Institutes of Health, Department of Health and Human Services. Beta-Carotene and Retinol Efficacy Trial (CARET): CARET is funded by the National Cancer Institute, National Institutes of Health through grants U01-CA063673, UM1-CA167462, and U01-CA167462 CLUE II: Campaign against Cancer and Heart Disease (CLUE II): Funding for the CLUE II cohort was provided by grants from the National Cancer Institute (U01-CA-086308) and the National Institute on Aging (U01-AG-18033). Cancer data were provided by the Maryland Cancer Registry, Center for Cancer Prevention and Control, Maryland Department of Health, with funding from the State of

Cancer Epidemiology, Nutritional Epidemioloy, German Cancer Research Center, Heidelberg, Germany, **25** Healthy Longevity Translational Research Programme, Yong Loo Lin School of Medicine, National University of Singapore, Singapore, Singapore, **26** Singapore Institute for Clinical Sciences, Agency for Science Technology and Research, Singapore, Singapore, **27** Department of Computational and Quantitative Medicine, Beckman Research Institute, City of Hope, Duarte, California, United States of America, **28** Department of Medicine Solna, Karolinska Institutet, Stockholm, Sweden, **29** Center for Obesity, Academic Specialist Center, Stockholm, Sweden, **30** Department of Food Science and Nutrition, The Hong Kong Polytechnic University, Hong Kong SAR, China, **31** Department of Public Health and Welfare, Finnish Institute for Health and Welfare, Helsinki, Finland, **32** Department of Population Science, American Cancer Society, Atlanta, Georgia, United States of America, **33** Dalla Lana School of Public Health, University of Toronto, Toronto, Ontario, Canada, **34** Precision Medicine, School of Clinical Sciences at Monash Health, Monash University, Clayton, Victoria, Australia, **35** Discovery Sciences, American Cancer Society, Atlanta, Georgia, United States of America, **36** Department of Epidemiology, Johns Hopkins Bloomberg School of Public Health, Baltimore, Maryland, United States of America, **37** Sidney Kimmel Comprehensive Cancer Center at Johns Hopkins, Baltimore, Maryland, United States of America, **38** Department of Laboratory Medicine and Pathology, University of Minnesota Medical School, and the University of Minnesota Masonic Cancer Center, Minneapolis, Minnesota, United States of America, **39** Department of Exercise and Nutrition Sciences, Milken Institute School of Public Health, George Washington University, Washington District of Columbia, United States of America, **40** Department of Epidemiology and Population Health, Albert Einstein College of Medicine, Bronx, New York, New York, United States of America, **41** Unit of Cancer Epidemiology, Città della Salute e della Scienza University-Hospital and Center for Cancer Prevention, Turin, Italy, **42** Department of Psychiatry, Icahn School of Medicine at Mount Sinai, New York, New York, United States of America, **43** Seaver Center for Autism Research and Treatment, Icahn School of Medicine at Mount Sinai, New York, New York, United States of America, **44** Division of Cohort Research, National Cancer Center Institute for Cancer Control, Tokyo, Japan, **45** IQVIA, Global Database Studies, Amsterdam, The Netherlands, **46** University of Iowa, Iowa City, Iowa, United States of America, **47** Mel & Enid Zuckerman College of Public Health, University of Arizona Cancer Center, Tucson, Arizona, United States of America, **48** Danish Cancer Institute, Diet, Cancer and Health, Copenhagen, Denmark, **49** Department of Epidemiology, Maastricht University, Maastricht, The Netherlands, **50** University of Pittsburgh Medical Center (UPMC) Hillman Cancer Center, Pittsburgh, Pennsylvania, United States of America, **51** Fred Hutchinson Cancer Research Center, Seattle, Washington, United States of America, **52** Department of Epidemiology, School of Public Health, University of Pittsburgh, Pittsburgh, Pennsylvania, United States of America

‡ SAS and PF are joint last authors on this work.
* ferrarip@iarc.who.int

## Abstract

### Background

Alcohol is a known carcinogen, yet the evidence for an association with pancreatic cancer risk is considered as limited or inconclusive by international expert panels. We examined the association between alcohol intake and pancreatic cancer risk in a large consortium of prospective studies.

### Methods and findings

Population-based individual-level data was pooled from 30 cohorts across four continents, including Asia, Australia, Europe, and North America. A total of 2,494,432 participants without cancer at baseline (62% women, 84% European ancestries, 70% alcohol drinkers [alcohol intake ≥ 0.1 g/day], 47% never smokers) were recruited between 1980 and 2013 at the median age of 57 years and 10,067 incident pancreatic cancer cases were recorded. In age- and sex-stratified Cox proportional hazards

Maryland and the Maryland Cigarette Restitution Fund. The collection and availability of cancer registry data are also supported by the Cooperative Agreement NU58DP007114, funded by the Centers for Disease Control and Prevention. Its contents are solely the responsibility of the authors and do not necessarily represent the official views of the Centers for Disease Control and Prevention or the Department of Health and Human Services. Canadian National Breast Screening Study (CNBSS): CNBSS was supported by the Canadian Cancer Society, the Department of National Health and Welfare, the National Cancer Institute of Canada, the Alberta Heritage Fund for Cancer Research, the Manitoba Health Services Commission, the Medical Research Council of Canada, the Ministry of Health and Social Services of Québec, the Nova Scotia Department of Health and the Ontario Ministry of Health. Cohort of Swedish Men (COSM): COSM is part of the Swedish Infrastructure for Medical Population-based Life-course and Environmental Research (SIMPLER), which receives funding through the Swedish Research Council under the grant no 2017-00644 and 2021-00160. Cancer Prevention Study-II Nutrition Cohort (CPS II): The American Cancer Society funds the creation, maintenance, and updating of the Cancer Prevention Study-II Cohort. California Teachers Study (CTS): CTS was supported by the National Cancer Institute of the National Institutes of Health under award number U01-CA199277; P30-CA033572; P30-CA023100; UM1-CA164917; and R01-CA077398. The content is solely the responsibility of the authors and does not necessarily represent the official views of the National Cancer Institute or the National Institutes of Health. European Investigation into Cancer and Nutrition (EPIC): The coordination of EPIC is financially supported by International Agency for Research on Cancer (IARC) and also by the Department of Epidemiology and Biostatistics, School of Public Health, Imperial College London (United Kingdom), which has additional infrastructure support provided by the National Institute for Health and Care Research (NIHR) Imperial Biomedical Research Centre (BRC). The national cohorts are supported by the Danish Cancer Society (Denmark); Ligue Contre le Cancer, Institut Gustave Roussy, Mutuelle Générale de l'Education Nationale, and Institut National de la Santé et de la Recherche Médicale (INSERM) (France); German Cancer Aid, German Cancer

models adjusted for smoking history, diabetes status, body mass index, height, education, race and ethnicity, and physical activity, pancreatic cancer hazard ratios (HR) and 95% confidence intervals (CI) were estimated for categories of alcohol intake and in continuous for a 10 g/day increase. Potential heterogeneity by sex, smoking status, geographic regions, and type of alcoholic beverage was investigated. Alcohol intake was positively associated with pancreatic cancer risk, with $HR_{30\text{-to-}<60\text{ g/day}}$ and $HR_{\geq 60\text{ g/day}}$ equal to 1.12 (95% CI [1.03,1.21]) and 1.32 (95% CI [1.18,1.47]), respectively, compared to intake of 0.1 to <5 g/day. A 10 g/day increment of alcohol intake was associated with a 3% increased pancreatic cancer risk overall (HR: 1.03; 95% CI [1.02,1.04]; $p_{value} < 0.001$) and among never smokers (HR: 1.03; 95% CI [1.01,1.06]; $p_{value} = 0.006$), with no evidence of heterogeneity by sex ($p_{heterogeneity} = 0.274$) or smoking status ($p_{heterogeneity} = 0.624$). Associations were consistent in Europe–Australia ($HR_{10\text{ g/day}} = 1.03$, 95% CI [1.00,1.05]; $p_{value} = 0.042$) and North America ($HR_{10\text{ g/day}} = 1.03$, 95% CI [1.02,1.05]; $p_{value} < 0.001$), while no association was observed in cohorts from Asia ($HR_{10\text{ g/day}} = 1.00$, 95% CI [0.96,1.03]; $p_{value} = 0.800$; $p_{heterogeneity} = 0.003$). Positive associations with pancreatic cancer risk were found for alcohol intake from beer ($HR_{10\text{ g/day}} = 1.02$, 95% CI [1.00,1.04]; $p_{value} = 0.015$) and spirits/liquor ($HR_{10\text{ g/day}} = 1.04$, 95% CI [1.03,1.06]; $p_{value} < 0.001$), but not wine ($HR_{10\text{ g/day}} = 1.00$, 95% CI [0.98,1.03]; $p_{value} = 0.827$). The differential associations across geographic regions and types of alcoholic beverages might reflect differences in drinking habits and deserve more investigations.

## Conclusions

Findings from this large-scale pooled analysis support a modest positive association between alcohol intake and pancreatic cancer risk, irrespective of sex and smoking status. Associations were particularly evident for baseline alcohol intake of at least 15 g/day in women and 30 g/day in men.

## Author summary

### Why was this study done?

- Alcohol consumption is a known carcinogen, yet the evidence for its association with the risk of pancreatic cancer was evaluated as inconclusive by international experts' panels

- Previous prospective investigations suggested a harmful role of alcohol in relation to pancreatic cancer development, particularly for alcohol intakes greater than 30 g/day, corresponding to daily intake of about 2 standard drinks of either beer, wine or liquors/spirits

Research Center (DKFZ), German Institute of Human Nutrition Potsdam-Rehbruecke (DIfE), and Federal Ministry of Education and Research (BMBF) (Germany); Associazione Italiana per la Ricerca sul Cancro (AIRC-Italy), Compagnia di San Paolo, and National Research Council (Italy); Dutch Ministry of Public Health, Welfare and Sports (VWS), Netherlands Cancer Registry (NKR), LK Research Funds, Dutch Prevention Funds, Zorg Onderzoek Nederland (ZON), World Cancer Research Fund (WCRF), and Statistics Netherlands (The Netherlands); Health Research Fund (FIS)—Instituto de Salud Carlos III (ISCIII), Regional Governments of Andalucia, Asturias, Basque Country, Murcia and Navarra, and the Catalan Institute of Oncology—ICO (Spain); Swedish Cancer Society, Swedish Research Council, and County Councils of Skane and Vasterbotten (Sweden); and Cancer Research UK (14136 and C8221/A29017) and Medical Research Council (grant numbers 1000143 and MR/M012190/1) (United Kingdom). Generations Study (GS): GS is supported by Breast Cancer Now and the National Institute for Health and Care Research (NIHR) Biomedical Research Centre at The Royal Marsden NHS Foundation Trust and the Institute of Cancer Research, London. The views expressed are those of the author(s) and not necessarily those of the NIHR or the Department of Health and Social Care. Health Professionals Follow-up Study (HPFS): HPFS was supported by grants from the National Institutes of Health (U01 CA167552). The study protocol was approved by the institutional review boards of the Brigham and Women's Hospital and Harvard T.H. Chan School of Public Health, and those of participating registries as required. Central registries may also be supported by state agencies, universities, and cancer centers. Participating central cancer registries include the following: Alabama, Alaska, Arizona, Arkansas, California, Delaware, Colorado, Connecticut, Florida, Georgia, Hawaii, Idaho, Indiana, Iowa, Kentucky, Louisiana, Maine, Maryland, Massachusetts, Michigan, Mississippi, Montana, Nebraska, Nevada, New Hampshire, New Jersey, New Mexico, New York, North Carolina, North Dakota, Ohio, Oklahoma, Oregon, Pennsylvania, Puerto Rico, Rhode Island, Seattle SEER Registry, South Carolina, Tennessee, Texas, Utah, Virginia, West Virginia, Wyoming. The content is solely the responsibility of the authors and does not necessarily represent the official views of the National

- The evaluation of the association among never smokers, women and alcohol subtypes was limited by the size of previous studies

- In this work, we leveraged data from a large international consortium of prospective studies to comprehensively examine the association between alcohol intake assessed at recruitment and pancreatic cancer risk

## What did the researchers do and find?

- Individual-level data from 30 cohorts from four continents were pooled and harmonized. About 2 million participants were included, and 10,067 developed pancreatic cancer over a median follow-up time of 16 years

- Each 10 g/day increase in alcohol intake was associated with a 3% increase in pancreatic cancer risk. In women, compared to weak intake (0.1–5 g/day), alcohol intake of 15–30 g/day was associated with a 12% increased risk, while in men alcohol intake of 30–60 and more than 60 g/day were associated with a 15% and 36% increased risk, respectively

- Sensitivity analyses indicated absence of heterogeneity by sex and smoking status, whereas evaluations by geographic region and alcohol subtypes showed null associations among participants from Asian cohorts and for alcohol intake from wine

## What do these findings mean?

- These findings support a modest positive association of alcohol intake with pancreatic cancer risk, irrespective of sex and smoking status. Associations were apparent for alcohol intake of at least 15 g/day in women and 30 g/day in men

- This observational study examined alcohol intake evaluated at a single time point during mid-to-late adulthood and included a limited number of Asian cohorts. Further research is needed to evaluate the role of lifetime alcohol consumption on pancreatic cancer risk as well as the impact of specific drinking patterns, including binge drinking.

## Introduction

Over the last decade pancreatic cancer has emerged as a major public health concern. Although ranked as the 12th most common cancer, it is often diagnosed at advanced stages and is highly fatal [1]. In 2022, pancreatic cancer accounted for 5% of cancer-related deaths worldwide [1]. Its incidence and mortality rates were 4–5 times higher in Europe, North America, Australia/New Zealand and Eastern Asia than in other regions, and no substantial improvement in survival has been observed in recent years [1–3]. Due to population growth, ageing and changes in the prevalence of potentially relevant lifestyle factors worldwide, the pancreatic cancer burden is expected to continue to rise [4]. While several risk factors for pancreatic cancer have

Institutes of Health. Iowa Women's Health Study (IWHS): IWHS was supported by a grant from the National Cancer Institute (NCI) of the United States (grant: R01 CA39742). Japan Public Health Center-based Prospective Study (JPHC) I and II: JPHC I and 2 were supported by National Cancer Center Research and Development Fund (23-A-31[toku], 26-A-2, 29-A-4, 2020-J-4, 2023-J-04) (since 2011) and a Grant-in-Aid for Cancer Research from the Ministry of Health, Labour and Welfare of Japan (from 1989 to 2010). Melbourne Collaborative Cohort Study (MCCS): MCCS recruitment was funded by VicHealth and Cancer Council Victoria. The MCCS was further augmented by Australian National Health and Medical Research Council grants 209057, 396414 and 1074383 and by infrastructure provided by Cancer Council Victoria. Cases and their vital status were ascertained through the Victorian Cancer Registry and the Australian Institute of Health and Welfare, including the Australian Cancer Database. Multiethnic Cohort Study (MEC): MEC is supported by NIH/NCI grant U01CA164973. Nurses' Health Study (NHS): NHS was supported by the National Institutes of Health (UM1 CA186107 and P01 CA87969). The study protocol was approved by the institutional review boards of the Brigham and Women's Hospital and Harvard T.H. Chan School of Public Health, and those of participating registries as required. Central registries may also be supported by state agencies, universities, and cancer centers. Participating central cancer registries include the following: Alabama, Alaska, Arizona, Arkansas, California, Delaware, Colorado, Connecticut, Florida, Georgia, Hawaii, Idaho, Indiana, Iowa, Kentucky, Louisiana, Maine, Maryland, Massachusetts, Michigan, Mississippi, Montana, Nebraska, Nevada, New Hampshire, New Jersey, New Mexico, New York, North Carolina, North Dakota, Ohio, Oklahoma, Oregon, Pennsylvania, Puerto Rico, Rhode Island, Seattle SEER Registry, South Carolina, Tennessee, Texas, Utah, Virginia, West Virginia, Wyoming. The content is solely the responsibility of the authors and does not necessarily represent the official views of the National Institutes of Health. Nurses' Health Study II (NHS II): NHSII was supported by the National Institutes of Health (U01 CA176726 and U01 HL145386). The study protocol was approved by the institutional review boards of the Brigham and Women's Hospital and Harvard T.H. Chan School of Public Health, and those of

been identified, including tobacco smoking, excess body fatness, chronic pancreatitis and diabetes mellitus [5,6], its aetiology remains poorly understood.

Although alcohol consumption was classified as a group 1-carcinogen by the International Agency for Research on Cancer (IARC), the evidence for an association with pancreatic cancer was judged to be sparse and inconsistent. Similarly, international expert panels from the World Cancer Research Fund/American Institute for Cancer Research considered the evidence of an association to be only suggestive [6,7]. As case–control studies are widely subject to recall bias [8–10], evaluation of the evidence from individual studies using a prospective design was prioritized. A prior evaluation of observational prospective data from 14 cohorts in the Pooling Project of Prospective Studies of Diet and Cancer (DCPP), showed a positive association, with an estimated 22% increased pancreatic cancer risk among participants with alcohol intake of at least 30 g of ethanol per day (g/day) (equivalent to 2 United-States [US] standard alcoholic drinks/day) as compared to non-drinkers [11]. However, since alcohol is often used jointly with tobacco, it has been suggested that the association could be confounded by smoking habits [7]. Large-scale American, European and Japanese cohorts showed limited heterogeneity of the alcohol and pancreatic cancer risk association by smoking status, while the relationships among never smokers were inconsistent [11–15], likely due to the small number of pancreatic cancer cases. In addition, previous findings showed inconsistent associations by type of alcoholic beverages [9,11,12,14] and across geographic regions [10,16,17].

In this study, we extended the prior analyses in the DCPP [11] and examined the association between alcohol consumption and pancreatic cancer risk in 30 prospective cohort studies spanning four continents. Individual-level data were pooled to evaluate the overall association, as well as by sex, smoking status, geographic region, and alcohol intake from specific beverage types.

## Materials and methods

### Study sample

A total of 30 prospective studies were included from the Pooling Project on Alcohol and Cancer (PPAC), an international consortium conducted within the DCPP (Table 1). The following inclusion criteria were pre-established by the DCPP to maximize the quality and the comparability across studies [18]: (1) prospective design, (2) at least one publication on diet and cancer, (3) long-term comprehensive dietary assessment method sufficient to calculate intakes of most nutrients including total energy, and (4) validation study of the dietary assessment method used in the study or a closely related instrument. Additional inclusion criteria for this project were that (5) alcohol intake in grams of ethanol per day was assessed, (6) that sex-specific sub-cohorts had over 10% alcohol drinkers (alcohol intake ≥ 0.1 g/day) at recruitment and (7) a minimum of 50 incident pancreatic cancer cases were documented during follow-up. Twenty-six cohorts were identified meeting the inclusion criteria and agreed to participate in the project. As part of the PPAC, four additional cohort studies were included that did not meet the inclusion criteria (2) to (4), but met the remaining inclusion criteria [19–21].

participating registries as required. The authors would like to acknowledge the contribution to this study from central cancer registries supported through the Centers for Disease Control and Prevention's National Program of Cancer Registries (NPCR) and/or the National Cancer Institute's Surveillance, Epidemiology, and End Results (SEER) Program. Central registries may also be supported by state agencies, universities, and cancer centers. Participating central cancer registries include the following: Alabama, Alaska, Arizona, Arkansas, California, Delaware, Colorado, Connecticut, Florida, Georgia, Hawaii, Idaho, Indiana, Iowa, Kentucky, Louisiana, Maine, Maryland, Massachusetts, Michigan, Mississippi, Montana, Nebraska, Nevada, New Hampshire, New Jersey, New Mexico, New York, North Carolina, North Dakota, Ohio, Oklahoma, Oregon, Pennsylvania, Puerto Rico, Rhode Island, Seattle SEER Registry, South Carolina, Tennessee, Texas, Utah, Virginia, West Virginia, Wyoming. The content is solely the responsibility of the authors and does not necessarily represent the official views of the National Institutes of Health. NIH-AARP Diet and Health Study (NIH-AARP): NIH-AARP was supported by the Intramural Research Program, Division of Cancer Epidemiology and Genetics of the US. Cancer incidence data from the Atlanta metropolitan area were collected by the Georgia Center for Cancer Statistics, Department of Epidemiology, Rollins School of Public Health, Emory University, Atlanta, Georgia. Cancer incidence data from California were collected by the California Cancer Registry, California Department of Public Health's Cancer Surveillance and Research Branch, Sacramento, California. Cancer incidence data from the Detroit metropolitan area were collected by the Michigan Cancer Surveillance Program, Community Health Administration, Lansing, Michigan. The Florida cancer incidence data used in this report were collected by the Florida Cancer Data System (Miami, Florida) under contract with the Florida Department of Health, Tallahassee, Florida. The views expressed herein are solely those of the authors and do not necessarily reflect those of the FCDC or FDOH. Cancer incidence data from Louisiana were collected by the Louisiana Tumor Registry, Louisiana State University Health Sciences Center School of Public Health, New Orleans, Louisiana. Cancer incidence data from New Jersey were collected by the New Jersey State Cancer Registry, The Rutgers Cancer Institute

## Ethic statement

Individual-level data from participating cohorts to the DCPP were centralized and harmonized at the Harvard T.H. Chan School of Public Health [11,12,14,15,19–43]. Details on harmonization are described in sections on exposure assessment, outcome assessment, and statistical analysis. Institutional review board approvals were received for each cohort, the consortium and participating cancer registries as required board (Table B in S1 File), and participants provided formal written informed consent before they completed questionnaires at baseline. Participants or the public were not involved in the design and the conduct of this study.

## Exposure assessment

The primary exposure of interest was self-reported alcohol intake in grams of ethanol per day. Data on alcohol intake and the baseline risk factors have been harmonized in the 30 studies. Most studies assessed alcohol intake using a food frequency questionnaire (FFQ) or a diet history that was tailored for their particular study and was designed to be comprehensive enough to estimate energy intake over a long period of time, generally the past year [18]. For the four cohorts that did not collect comprehensive dietary data, alcohol intake was estimated using a lifestyle questionnaire. Daily amount of alcohol in grams of ethanol was calculated from the baseline study-specific questionnaires based on the frequency of consumption, the number of drinks and the alcohol content of the alcoholic beverages consumed. All studies had a measure of total alcohol intake at enrolment calculated as the sum of the beverage-specific intakes, namely beer, wine and spirits/liquor. Information on the type of alcoholic beverage was not available in the New York State Cohort (NYSC), thus only total alcohol intake was used from this cohort. Measurement of alcohol intake by self-reported questionnaire was validated and correlation between the FFQ/diet history and 24-hour dietary recall measurements ranged from 0.74 to 0.99 [44–53].

For the 30 studies, the baseline non-dietary risk factor data included age at alcohol assessment, year of baseline questionnaire, sex, country, smoking habits, weight, height, race, ethnicity, education, physical activity, and prevalent diabetes (of any type).

## Outcome assessment

First primary incident pancreatic cancer cases, defined by the International Classification of Diseases (ICD) code 157 (9th edition [54]) or C25 (10th edition [55]), were ascertained by self-report with subsequent medical record review [42], cancer registry linkage [31,32,34,35,38,39,56], or both approaches [27–29,33]. Some studies additionally obtained information from death registries [27,28,32–35,38,39,42]. We excluded endocrine (ICD-9 code 157.4 and ICD-10 code C25.4) and lymphoproliferative tumours (ICD-Oncology 3rd edition [57] codes: 9251, 9560, 9590, 9591, 9680, 9691, 9695, 9950). The final data included 10,067 invasive pancreatic cancer cases, from which 9,668 cases (96%) were histologically confirmed (with ICD-O codes other than 9990, 9999).

of New Jersey, New Brunswick, New Jersey. Netherlands Cohort Study (NLCS): The NLCS was supported by grants from the Dutch Cancer Society and World Cancer Research Fund. New York State Cohort (NYSC): No source of funding to declare. Prostate, Lung, Colorectal, and Ovarian Cancer Screening Trial (PLCO): PLCO was supported by the Intramural Research Program of the Division of Cancer Epidemiology and Genetics, and contracts from the Division of Cancer Prevention, National Cancer Institute (NCI), NIH, DHHS. Cancer incidence data have been provided by the Colorado Central Cancer Registry, District of Columbia Cancer Registry, Georgia Cancer Registry, Hawaii Cancer Registry, Cancer Data Registry of Idaho, Minnesota Cancer Surveillance System, Missouri Cancer Registry, Nevada Central Cancer Registry, Pennsylvania Cancer Registry, Texas Cancer Registry, Virginia Cancer Registry, and Wisconsin Cancer Reporting System. All are supported in part by funds from the Center for Disease Control and Prevention, National Program for Central Registries, local states or by the National Cancer Institute, Surveillance, Epidemiology, and End Results program. The results reported here, and the conclusions derived are the sole responsibility of the authors. Singapore Chinese Health Study (SCHS): SCHS was supported by the National Institutes of Health (NIH) of the United States (grants # R01 CA144034 and UM1 CA182876; PI: J-M.Y.) and the Singapore Ministry of Health's National Medical Research Council under its Strategic Cohorts Funding (SCHS; PI: W-P.K.). Shanghai Cohort Study (SCS): SCS was supported by the National Institutes of Health (NIH) of the United States (grants # R01 CA144034 and UM1 CA182876; PI: J-M.Y.) Swedish Mammography Cohort (SMC): SMC is part of the Swedish Infrastructure for Medical Population-based Life-course and Environmental Research (SIMPLER), which receives funding through the Swedish Research Council under the grant no 2017-00644 and 2021-00160. Shanghai Men's Health Study (SMHS): SMHS is supported by National Institutes of Health (NIH) of the United States (grant UM1CA173640). Swedish National March Cohort (SNMC): No source of funding to declare. VITamins and Lifestyle Study (VITAL): No source of funding to declare. Women's Health Initiative (WHI): The WHI program is funded by the National Cancer Heart, Lung, and Blood Institute, National Institutes of Health, U.S. Department of Health

### Statistical analysis

In addition to applying predefined exclusions for each cohort, we excluded participants with a prior cancer diagnosis other than non-melanoma skin cancer at baseline, $\log_e$-transformed energy intakes beyond three standard deviations of the study- and sex-specific $\log_e$-transformed mean energy intake (for all cohorts with comprehensive dietary data), with missing data on alcohol intake, or alcohol intakes exceeding 200 g/day. The Netherlands Cohort Study (NLCS) was analysed as a case-cohort study per its study design [56].

The association between alcohol intake and pancreatic cancer risk was evaluated using multivariable Cox proportional hazards models on aggregated individual-level data from each study into a unique dataset, to estimate hazard ratios (HR) and 95% confidence intervals (CI). The primary time scale variable was the follow-up time in years from age at baseline until the age at cancer diagnosis (only participants for whom pancreatic cancer was their first primary cancer were included as cases), death, or administrative end of follow-up, whichever occurred first. The baseline hazard was stratified by age at recruitment (in 1-year categories), year of baseline questionnaire completion (in 1-calendar-year categories), study, country (in the European Prospective Investigation into Cancer and Nutrition [EPIC] cohort), and sex (in models combining men and women).

Alcohol intake was modelled in categories as: <0.1, 0.1 to <5, 5 to <15, 15 to <30, 30 to <60 and ≥60 g/day and in continuous for a 10 g/day increase. The category 0.1 to <5 g/day was used as the reference category as non-drinkers at baseline (<0.1 g/day) may include former drinkers. In women, the two highest categories of alcohol intake were collapsed into a ≥30 g/day group (owing to 49 cases in the ≥60 g/day category). The covariates were categorized similarly across studies. Models were adjusted for smoking status (never, former, current, unknown: 2%), smoking duration in past and current smokers (in years, coded as 0 for never smokers and missing [17%]), smoking intensity in current smokers (in number of cigarettes/day, coded as 0 for never and former smokers and missing [3%]), time since smoking cessation in past smokers (in years, coded as 0 for never and current smokers and missing [4%]), prevalent diabetes status (yes, no, unknown: 6%), body mass index (BMI, continuous, kg/m$^2$, unknown: 4%), height (continuous, centimetres, unknown: 1%), education level (<high school, high school, >high school, unknown: 6%), self-identified race and ethnicity (African American, Asian, White, Hispanic, Other, unknown: 3%), and physical activity (low, medium, high, unknown: 17%). Missing values for covariates were modelled with missing indicator variables. Participants from European studies that did not collect information about race or ethnicity were assigned to the "White" category. Analyses on the continuous scale were further adjusted for an indicator variable for alcohol drinking status (0: non-drinkers [<0.1 g/day], 1: drinkers [≥0.1 g/day]) to express alcohol intake in continuous among drinkers. The inclusion of energy intake did not alter the magnitude of the HR estimates and was not included in the final multivariable models.

Tests for statistical significance of pancreatic cancer HRs related to alcohol intake in categories were performed with $p$-values ($p_{Wald}$) comparing the Wald test statistics

and Human Services through contracts 75N92021D00001, 75N92021D00002, 75N92021D00003, 75N92021D00004, 75N92021D00005. Women's Lifestyle and Health Study (WLHS): No source of funding to declare.

**Competing interests:** I have read the journal's policy and the authors of this manuscript have the following competing interests: MS is employed at IQVIA, a contract research company, and has no financial stake in the results of the current study. WZ and SCM are Academic Editors on PLOS Medicine's editorial board.

**Abbreviations:** BMI, body mass index; DCPP, Pooling Project of Prospective Studies of Diet and Cancer; HR, hazard ratio; CI, confidence interval; PPAC, Pooling Project on Alcohol and Cancer; ATBC, Alpha-Tocopherol, Beta-Carotene Cancer Prevention Study; BCDDP, breast cancer detection demonstration project follow-up study; CARET, beta-carotene and retinol efficacy trial; CLUE II, Clue II: Campaign against Cancer and Heart Disease; CNBSS, Canadian National Breast Screening Study; COSM, Cohort of Swedish Men; CPS II, Cancer Prevention Study II Nutrition Cohort; CTS, California Teachers Study; EPIC, European Prospective Investigation into Cancer and Nutrition; GS, Generations Study; HPFS, Health Professionals Follow-up Study; IWHS, Iowa Women's Health Study; JPHC I, Japan Public Health Center-based Prospective Study I; JPHC II, Japan Public Health Center-based Prospective Study II; MCCS, Melbourne Collaborative Cohort Study; MEC, Multiethnic Cohort Study; NHS, Nurses' Health Study; NHS II, Nurses' Health Study II; NIH-AARP, NIH-AARP Diet and Health Study; NLCS, Netherlands Cohort Study; NYSC, New York State Cohort; PLCO, Prostate, Lung, Colorectal, and Ovarian Cancer Screening Trial; SCHS, Singapore Chinese Health Study; SCS, Shanghai Cohort Study; SMC, Swedish Mammography Cohort; SMHS, Shanghai Men's Health Study; SNMC, Swedish National March Cohort; VITAL, VITamins and Lifestyle Study: Cohort Study of Dietary Supplements and Cancer Risk; WHI, Women's Health Initiative; WLHS, Women's Lifestyle and Health Study.

to a $X^2$ distribution with degrees of freedom equal to the number of alcohol categories minus one, not including the category of non-drinkers (<0.1 g/day). P-values for trend ($p_{trend}$) were obtained in models including alcohol intake as a continuous variable, also including the indicator for alcohol drinking status.

To assess potential departures from linearity in the association between alcohol intake and pancreatic cancer risk, multivariable adjusted restricted cubic spline models [58] were fitted with four internal knots placed at 5, 15, 30, and 60 g/day using 2.5 g/day as the reference, after excluding participants with alcohol intake greater than 100 g/day (resulted in exclusion of 5% of sex-combined participants). Models included the same list of covariates as detailed earlier for alcohol intake modelled in continuous. Nonlinearity was evaluated by comparing the difference in log-likelihood of models with linear term and fractional polynomials to a $X^2$ distribution with two degrees of freedoms.

We evaluated heterogeneity in the alcohol intake and pancreatic cancer risk association by study, sex, smoking status (never, former, current smokers), geographic region (Europe-Australia, North America, Asia), BMI (18.5 to <25, 25 to <30, ≥30 kg/m²]), prevalent diabetes status (yes, no), education level (≤high school, >high school), multi-vitamin use (yes, no, in North American cohorts), and follow-up time (<2, 2 to <5, 5 to <10, ≥10 years). Models included alcohol intake (in continuous), a categorical variable for the candidate effect modifier, and an interaction term between alcohol intake and the effect modifier. P-values for heterogeneity were obtained by comparing the log-likelihood of models with and without the interaction terms to a $X^2$ distribution with degrees of freedom equal to the number of categories of the effect modifier minus one. Participants with missing values on the candidate effect modifier were excluded in models evaluating that modifier and models were adjusted as previously described for alcohol intake modelled in continuous. In analyses by geographical region, the Australian cohort (Melbourne Collaborative Cohort Study [MCCS]) was combined with European cohorts as participants were mostly White and reported similar alcohol intake as European cohorts. Proportional hazards (PH) assumption was evaluated introducing a continuous time dependent variable modelled as the interaction between the log-transformed follow-up time and alcohol intake. The PH assumption was not rejected (p-value = 0.758).

The associations between alcohol intake from different alcoholic beverages (beer, wine and spirits/liquors) and pancreatic cancer risk were assessed in separate models, using the following categories: <0.1 g/day, 0.1 to <3 (reference), 3 to <10, 10 to <20, 20 to <40 and ≥40 g/day. These models were further adjusted for the sum of the alcohol intake from alcoholic beverages other than the one under evaluation.

Sensitivity analyses were performed to assess the robustness of the findings. To evaluate the effect of different adjustments of smoking variables on pancreatic cancer HRs related to alcohol, different models were compared: (1) no smoking covariates (only age, year of questionnaire return, and country [for EPIC only] were included as stratification variables), (2) adjustment for smoking status, (3) further adjustment for smoking duration, smoking intensity, and time since smoking cessation, (4) further adjustment for other covariates. To assess potential reverse causation, the association between alcohol use

Table 1. Characteristics of the cohort studies in the pooled analysis of alcohol intake and pancreatic cancer risk.

| Study[†] | Country/ Continent | Total number of participants[‡] | Year of baseline questionnaire | Age at baseline (years)[§] | Women[§] | Drinkers[§,¶] | Alcohol intake among drinkers (g/day)[§] | Never smokers[§,#] | Follow-up (years)[§] | Number cases | Age at diagnosis (years)[§] |
|---|---|---|---|---|---|---|---|---|---|---|---|
| ATBC | Finland | 26,815 | 1985–1988 | 57 (51, 65) | 0 (0%) | 23,768 (89%) | 13.0 (1.6, 45.7) | 0 (0%)* | 16.0 (4.4, 28.0) | 342 | 72 (62, 80) |
| BCDDP | USA | 42,140 | 1987–1989 | 60 (51, 72) | 42,140 (100%) | 20,587 (49%) | 3.5 (0.4, 20.1) | 23,736 (56%) | 10.3 (9.2, 10.9) | 54 | 68 (57, 83) |
| CARET | USA | 16,467 | 1985–1994 | 57 (50, 66) | 5,998 (36%) | 10,690 (65%) | 10.6 (1.0, 55.2) | 110 (1%) | 19.4 (5.1, 23.6) | 167 | 72 (62, 81) |
| CLUE II | USA | 7,812 | 1989 | 52 (30, 71) | 7,812 (100%)& | 2,543 (33%) | 2.6 (0.9, 16.5) | 4,924 (63%) | 17.6 (7.2, 23.5) | 50 | 71 (63, 87) |
| CNBSS | Canada | 48,778 | 1980–1985 | 48 (41, 56) | 48,778 (100%) | 37,374 (77%) | 6.6 (1.0, 27.7) | 24,659 (51%) | 22.1 (16.9, 24.4) | 198 | 66 (54, 75) |
| COSM | Sweden | 45,219 | 1998 | 59 (48, 74) | 0 (0%) | 41,411 (92%) | 9.0 (1.6, 23.6) | 16,139 (36%) | 19.0 (5.0, 19.0) | 206 | 72 (62, 81) |
| CPS II | USA | 140,491 | 1992–1993 | 63 (55, 71) | 74,603 (53%) | 82,133 (58%) | 6.7 (0.9, 35.7) | 62,194 (44%) | 19.8 (5.6, 21.4) | 623 | 74 (65, 83) |
| CTS | USA | 99,741 | 1995–1999 | 51 (34, 72) | 99,741 (100%) | 64,182 (64%) | 7.5 (3.2, 22.2) | 66,977 (67%) | 20.0 (8.0, 20.1) | 425 | 76 (60, 86) |
| EPIC | Europe* | 456,021 | 1991–2001 | 51 (39, 63) | 322,945 (71%) | 389,670 (85%) | 6.9 (0.7, 34.3) | 221,904 (49%) | 17.1 (9.9, 20.2) | 1,298 | 67 (55, 77) |
| GS | England | 104,713 | 2003–2013 | 47 (28, 64) | 104,713 (100%) | 82,762 (79%) | 14.0 (5.0, 36.0) | 67,380 (64%) | 9.0 (6.0, 10.0) | 56 | 68 (54, 76) |
| HPFS | USA | 47,766 | 1986–1987 | 54 (42, 68) | 0 (0%) | 36,458 (76%) | 9.6 (1.8, 36.9) | 21,308 (45%) | 25.9 (7.4, 30.7) | 297 | 72 (60, 84) |
| IWHS | USA | 34,578 | 1986 | 61 (56, 68) | 34,578 (100%) | 15,585 (45%) | 3.4 (0.9, 21.3) | 22,389 (65%) | 21.8 (6.2, 27.9) | 307 | 77 (66, 88) |
| JPHC I | Japan | 39,032 | 1995 | 55 (46, 63) | 20,792 (53%) | 17,418 (45%) | 23.0 (1.6, 72.0) | 26,051 (67%) | 18.0 (9.0, 18.0) | 225 | 68 (59, 77) |
| JPHC II | Japan | 49,183 | 1998–1999 | 59 (47, 71) | 26,044 (53%) | 21,177 (43%) | 23.0 (1.6, 74.6) | 29,344 (60%) | 15.0 (7.0, 15.0) | 244 | 73 (60, 82) |
| MCCS | Australia | 36,085 | 1990–1994 | 54 (42, 66) | 21,877 (61%) | 22,062 (61%) | 12.7 (1.1, 43.9) | 21,295 (59%) | 20.4 (9.6, 22.5) | 132 | 75 (61, 84) |
| MEC | USA | 36,821 | 1993–1997 | 57 (47, 71) | 19,692 (53%) | 23,653 (64%) | 9.7 (0.9, 47.6) | 14,321 (39%) | 18.6 (5.6, 20.3) | 201 | 72 (62, 82) |
| NHS | USA | 68,447 | 1986–1987 | 53 (43, 63) | 68,447 (100%) | 44,003 (64%) | 4.7 (0.9, 27.5) | 30,131 (44%) | 28.5 (10.2, 28.9) | 486 | 72 (61, 82) |
| NHS II | USA | 93,088 | 1991–1993 | 36 (30, 42) | 93,088 (100%) | 53,206 (57%) | 2.8 (0.9, 12.5) | 61,016 (66%) | 22.0 (21.0, 22.0) | 65 | 53 (45, 60) |
| NIH-AARP | USA | 490,722 | 1995–1997 | 62 (54, 68) | 199,267 (41%) | 368,230 (75%) | 4.5 (0.5, 37.3) | 173,045 (35%) | 15.5 (3.7, 15.8) | 2,694 | 72 (63, 79) |
| NLCS | NL | 120,852∞ | 1986 | 61 (56, 67) | 62,573 (52%) | —∞ (77%) | 8.7 (0.9, 32.1) | —∞ (35%) | 17.3 (4.0, 17.3) | 236 | 62 (56, 68) |
| NYSC | USA | 30,335 | 1980 | 60 (47, 74) | 0 (0%)& | 26,974 (89%) | 4.8 (0.2, 34.2) | 8,452 (28%) | 7.5 (5.5, 7.8) | 92 | 69 (58, 81) |
| PLCO | USA | 101,308 | 1993–2001 | 65 (58, 74) | 52,210 (52%) | 73,376 (72%) | 4.1 (0.6, 28.6) | 48,456 (48%) | 9.2 (4.5, 10.6) | 375 | 73 (64, 81) |
| SCHS | Singapore | 27,175 | 1993–1999 | 56 (46, 68) | 0 (0%)& | 8,549 (31%) | 4.4 (0.4, 28.1) | 11,482 (42%) | 14.3 (5.7, 17.2) | 68 | 69 (57, 79) |
| SCS | China | 16,339 | 1986–1989 | 55 (47, 62) | 0 (0%) | 6,921 (42%) | 21.5 (4.2, 64.5) | 7,029 (43%) | 25.1 (8.3, 29.2) | 160 | 71 (62, 82) |
| SMC | Sweden | 34,982 | 1997 | 60 (51, 76) | 34,982 (100%) | 28,695 (82%) | 3.2 (0.4, 10.2) | 18,641 (53%) | 19.3 (6.8, 19.3) | 164 | 74 (64, 84) |
| SMHS | China | 61,065 | 2001–2006 | 53 (43, 70) | 0 (0%) | 19,760 (32%) | 26.0 (8.9, 69.8) | 18,592 (30%) | 12.2 (8.6, 14.2) | 207 | 70 (54, 80) |
| SNMC | Sweden | 25,049 | 1997 | 50 (27, 69) | 25,049 (100%)& | 21,745 (87%) | 6.3 (0.8, 23.1) | 15,396 (61%) | 19.2 (10.6, 19.2) | 62 | 70 (58, 82) |
| VITAL | USA | 60,324 | 2000–2002 | 60 (52, 72) | 30,156 (50%) | 38,817 (64%) | 6.8 (0.8, 32.5) | 28,857 (48%) | 9.9 (3.9, 10.9) | 190 | 71 (62, 79) |
| WHI | USA | 85,545 | 1992–1995 | 63 (53, 73) | 85,545 (100%) | 49,509 (58%) | 4.6 (0.8, 23.6) | 43,130 (50%) | 18.5 (7.0, 20.9) | 386 | 75 (64, 85) |

*(Continued)*

Table 1. (Continued)

| Study[†] | Country/ Continent | Total number of participants[‡] | Year of baseline questionnaire | Age at baseline (years)[§] | Women[§] | Drinkers[§,¶] | Alcohol intake among drinkers (g/day)[§] | Never smokers[§,#] | Follow-up (years)[§] | Number cases | Age at diagnosis (years)[§] |
|---|---|---|---|---|---|---|---|---|---|---|---|
| WLHS | Sweden | 47,539 | 1991–1992 | 40 (32, 48) | 47,539 (100%) | 40,668 (86%) | 2.9 (0.5, 9.0) | 19,337 (41%) | 21.3 (20.4, 21.3) | 57 | 60 (50, 69) |
| Total | | 2,494,432 | 1980–2013 | 57 (40, 69) | 1,528,569 (62%) | –[∞] (70%) | 6.6 (0.8, 34.3) | –[∞] (47%) | 15.6 (6.0, 21.6) | 10,067 | 71 (60, 81) |

[†]Study abbreviations: ATBC: Alpha-Tocopherol Beta-Carotene Cancer Prevention Study [27]; BCDDP: Breast Cancer Detection Demonstration Project Follow-Up Study [28]; CARET: Beta-Carotene and Retinol Efficacy Trial [29]; CLUE II: Campaign against Cancer and Heart Disease [30]; CNBSS: Canadian National Breast Screening Study [31]; COSM: Cohort of Swedish Men [32]; CPS II: Cancer Prevention Study II Nutrition Cohort [33]; CTS: California Teachers Study [34]; EPIC: European Prospective Investigation into Cancer and Nutrition [14]; GS: Generations Study [19]; HPFS: Health Professionals Follow-up Study [42]; IWHS: Iowa Women's Health Study [35]; JPHC I: Japan Public Health Center-based Prospective Study I [15]; JPHC II: Japan Public Health Center-based Prospective Study II [15]; MCCS: Melbourne Collaborative Cohort Study [36]; MEC: Multiethnic Cohort Study [37]; NHS: Nurses' Health Study [42]; NHS II: Nurses' Health Study II [42]; NIH-AARP: NIH-AARP Diet and Health Study [12]; NL: The Netherlands NLCS: the Netherlands Cohort Study [43]; NYSC: New York State Cohort [38]; PLCO: Prostate, Lung, Colorectal, and Ovarian Cancer Screening Trial [40]; SCHS: Singapore Chinese Health Study [22]; SCS: Shanghai Cohort Study [20]; SMC: Swedish Mammography Cohort [32]; SMHS: Shanghai Men's Health Study [24]; SNMC: Swedish National March Cohort [21]; VITAL: VITamins and Lifestyle Study: Cohort Study of Dietary Supplements and Cancer Risk [25]; WHI: Women's Health Initiative [26]; WLHS: Women's Lifestyle and Health Study [41].

[‡]Cohort size reflects the size after application of study-specific exclusion criteria and further exclusion of participants with energy intakes beyond 3 standard-deviations of their $\log_e$-transformed study-specific mean energy intake (except for the four cohorts without dietary assessment), history of cancer diagnosis at baseline (except for nonmelanoma skin cancer), missing data on alcohol intake, or alcohol intakes exceeding 200 g/day. The Netherlands Cohort Study was analysed as a case-cohort study, and the above exclusions were not applied to its baseline cohort size.

[§]Median (10th, 90th percentiles) for continuous variables, number and proportion of cohort for categorical variables.

[¶]Drinkers are participants with a total alcohol intake ≥ 0.1 g/day at baseline.

[#]Never smokers are participants who reported having never smoked at baseline; and Men in CLUE II and SNMC were not included because the number of pancreatic cancer cases was lower than 50. Women in NYSC were not included because the number of pancreatic cancer cases was lower than 50. Women in SCHS were not included because the prevalence of drinking at baseline was lower than 10%.

[*]In ATBC only current smokers were recruited.

[∞]NLCS was analysed as a case-cohort study. Data are only available for the subcohort and pancreatic cancer cases. Therefore, the number of participants was not presented and the percentages are representative of the subcohort.

[♦]EPIC included participants from 9 European countries: Denmark, France, Germany, Italy, the Netherlands, Norway, Spain, Sweden, and United Kingdom.

**Table 2. Participant characteristics by category of alcohol intake by sex[‡].**

| | | Alcohol intake (g/day) | | | | | | Total |
|---|---|---|---|---|---|---|---|---|
| | | <0.1 | 0.1 to <5 | 5 to <15 | 15 to <30 | 30 to <60 | 60+ | |
| **WOMEN** | | | | | | | | |
| Participants | n (%) | 473,239 (32%) | 496,793 (34%) | 304,577 (21%) | 125,706 (9%) | 57,404 (4%) | 10,187 (1%) | 1,467,906 (100%) |
| Person-Years | N | 7,437,176 | 7,969,119 | 4,793,668 | 1,826,469 | 827,773 | 139,839 | 22,994,044 |
| Follow-up duration | Years | 16.2 (6.5, 22.0) | 16.5 (7.6, 22.0) | 17.2 (6.3, 21.8) | 15.6 (6.0, 21.3) | 15.6 (6.0, 21.4) | 15.4 (5.8, 21.0) | 16.4 (6.6, 22.0) |
| Number of PC cases | n (%) | 1,827 (35%) | 1,749 (34%) | 884 (17%) | 438 (8%) | 220 (4%) | 49 (1%) | 5,167 (100%) |
| Alcohol intake | g/day | 0 (0, 0) | 1.6 (0.4, 3.9) | 8.6 (5.5, 13.2) | 19.8 (15.6, 27.0) | 36.9 (31.0, 51.4) | 72.8 (62.0, 118.3) | 1.7 (0.0, 18.1) |
| Age at baseline | Years | 57 (39, 69) | 55 (37, 68) | 53 (35, 67) | 54 (37, 67) | 54 (39, 66) | 56 (42, 67) | 55 (37, 68) |
| Never smokers | n (%) | 317,164 (67%) | 267,836 (54%) | 154,265 (51%) | 55,530 (44%) | 19,437 (34%) | 2,644 (26%) | 816,876 (56%) |
| Diabetes | n (%) | 31,693 (7%) | 15,079 (3%) | 4,674 (2%) | 1,785 (1%) | 929 (2%) | 225 (2%) | 54,385 (4%) |
| Height | m | 1.6 (1.5, 1.7) | 1.6 (1.5, 1.7) | 1.6 (1.6, 1.7) | 1.6 (1.6, 1.7) | 1.6 (1.6, 1.7) | 1.6 (1.6, 1.7) | 1.6 (1.5, 1.7) |
| Body mass index | kg/m² | 25.0 (20.5, 33.5) | 24.6 (20.4, 31.8) | 23.8 (20.2, 29.9) | 23.8 (20.2, 29.4) | 23.9 (20.3, 29.9) | 24.4 (20.3, 31.0) | 24.6 (20.4, 31.8) |
| Education level ≥ high school | n (%) | 357,243 (75%) | 401,065 (81%) | 255,934 (84%) | 108,362 (86%) | 50,393 (88%) | 8,892 (87%) | 1,181,889 (81%) |
| Physical activity level ≥ medium | n (%) | 230,090 (49%) | 272,025 (55%) | 149,630 (49%) | 53,082 (42%) | 24,104 (42%) | 4,323 (42%) | 733,254 (50%) |
| *Geographical region* | | | | | | | | |
| Europe | n (%) | 96,353 (20%) | 194,858 (39%) | 151,357 (50%) | 62,677 (50%) | 28,244 (49%) | 3,649 (36%) | 537,138 (37%) |
| North America | n (%) | 338,317 (71%) | 297,517 (60%) | 151,040 (50%) | 62,045 (49%) | 28,684 (50%) | 6,329 (62%) | 883,932 (60%) |
| Asia-Pacific | n (%) | 38,569 (8%) | 4,418 (1%) | 2,180 (1%) | 984 (1%) | 476 (1%) | 209 (2%) | 46,836 (3%) |
| *Race* | | | | | | | | |
| European descent | n (%) | 377,141 (80%) | 449,320 (90%) | 277,611 (91%) | 114,598 (91%) | 52,161 (91%) | 8,967 (88%) | 1,279,798 (87%) |
| African descent | n (%) | 14,937 (3%) | 8,714 (2%) | 2,751 (1%) | 842 (1%) | 454 (1%) | 175 (2%) | 27,873 (2%) |
| Hispanic descent | n (%) | 6,335 (1%) | 5,013 (1%) | 2,466 (1%) | 629 (1%) | 215 (0%) | 55 (1%) | 14,713 (1%) |
| Asian descent | n (%) | 48,389 (10%) | 7,992 (2%) | 3,595 (1%) | 1,346 (1%) | 585 (1%) | 222 (2%) | 62,129 (4%) |
| Other/unknown | n (%) | 26,437 (6%) | 25,754 (5%) | 18,154 (6%) | 8,291 (7%) | 3,989 (7%) | 768 (8%) | 83,393 (6%) |
| **MEN** | | | | | | | | |
| Participants | n (%) | 229,293 (25%) | 226,897 (25%) | 181,953 (20%) | 132,303 (15%) | 93,345 (10%) | 45,732 (5%) | 909,523 (100%) |
| Person-Years | N | 2,993,148 | 2,955,696 | 2,554,297 | 1,813,728 | 1,290,708 | 592,021 | 12,199,598 |
| Follow-up duration | Years | 13.1 (4.4, 20.8) | 15.0 (4.4, 19.8) | 15.5 (5.1, 20.9) | 15.0 (5.1, 20.2) | 15.0 (5.1, 20.3) | 14.9 (4.3, 19.0) | 14.7 (4.7, 20.3) |
| Number of PC cases | n (%) | 1,204 (25%) | 1,152 (24%) | 922 (19%) | 712 (15%) | 576 (12%) | 334 (7%) | 4,900 (100%) |
| Alcohol intake | g/day | 0 (0, 0) | 2.0 (1.8, 0.5) | 9.5 (9.4, 5.7) | 21.3 (21.0, 16.0) | 42.4 (41.2, 32.2) | 89.0 (80.3, 63.3) | 4.9 (0, 41.8) |
| Age at baseline | Years | 60 (47, 70) | 60 (49, 69) | 59 (46, 69) | 59 (46, 69) | 58 (46, 68) | 59 (47, 68) | 59 (47, 69) |
| Never smokers | n (%) | 89,872 (39%) | 81,817 (36%) | 58,413 (32%) | 34,519 (26%) | 18,893 (20%) | 7,263 (16%) | 290,777 (32%) |
| Diabetes | n (%) | 24,520 (11%) | 17,541 (8%) | 8,146 (4%) | 5,147 (4%) | 3,825 (4%) | 2,360 (5%) | 61,539 (7%) |
| Height | m | 1.7 (1.6, 1.8) | 1.8 (1.7, 1.9) | 1.8 (1.7, 1.9) | 1.8 (1.7, 1.9) | 1.8 (1.6, 1.9) | 1.8 (1.7, 1.9) | 1.8 (1.7, 1.9) |
| Body mass index | kg/m² | 25.1 (20.9, 31.1) | 26.1 (22.3, 32) | 25.8 (22.1, 30.8) | 25.7 (22, 30.6) | 25.8 (21.8, 30.8) | 25.9 (21.6, 31.4) | 25.8 (21.8, 31.2) |
| Education level ≥ high school | n (%) | 157,205 (69%) | 179,472 (79%) | 137,554 (76%) | 98,823 (75%) | 66,183 (71%) | 30,859 (67%) | 670,096 (74%) |
| Physical activity level ≥ medium | n (%) | 102,364 (45%) | 121,537 (54%) | 102,975 (57%) | 74,381 (56%) | 50,328 (54%) | 24,921 (54%) | 476,506 (52%) |
| *Geographical region* | | | | | | | | |
| Europe | n (%) | 16,144 (7%) | 47,492 (21%) | 62,016 (34%) | 42,145 (32%) | 29,307 (31%) | 9,945 (22%) | 207,049 (23%) |
| North America | n (%) | 132,749 (58%) | 169,266 (75%) | 107,964 (59%) | 73,581 (56%) | 47,625 (51%) | 25,331 (55%) | 556,516 (61%) |
| Asia-Pacific | n (%) | 80,400 (35%) | 10,139 (4%) | 11,973 (7%) | 16,577 (13%) | 16,413 (18%) | 10,456 (23%) | 145,958 (16%) |

*(Continued)*

**Table 2.** (Continued)

| Race | | Alcohol intake (g/day) | | | | | | Total |
|---|---|---|---|---|---|---|---|---|
| | | <0.1 | 0.1 to <5 | 5 to <15 | 15 to <30 | 30 to <60 | 60+ | |
| European descent | n (%) | 137,921 (60%) | 204,654 (90%) | 163,757 (90%) | 112,186 (85%) | 74,871 (80%) | 34,152 (75%) | 727,541 (80%) |
| African descent | n (%) | 3,822 (2%) | 3,688 (2%) | 1,875 (1%) | 1,065 (1%) | 653 (1%) | 408 (1%) | 11,511 (1%) |
| Hispanic descent | n (%) | 1,506 (1%) | 2,663 (1%) | 1,443 (1%) | 841 (1%) | 461 (0%) | 269 (1%) | 7,183 (1%) |
| Asian descent | n (%) | 83,169 (36%) | 13,046 (6%) | 13,042 (7%) | 17,230 (13%) | 16,728 (18%) | 10,594 (23%) | 153,809 (17%) |
| Other/unknown | n (%) | 2,875 (1%) | 2,846 (1%) | 1,836 (1%) | 981 (1%) | 632 (1%) | 309 (1%) | 9,479 (1%) |

Abbreviations: PC = pancreatic cancer.

†Median and 10th–90th percentile range for continuous variables; numbers and sex-specific percentage for categorical variables.

and pancreatic cancer risk was re-evaluated after excluding the first 2 years of follow-up. The association was also re-examined after restricting the case definition to histologically confirmed pancreatic cancer cases. Among cohorts with information about past drinking (Cohort of Swedish Men [COSM], EPIC, Health Professionals Follow-up Study [HPFS], Melbourne Collaborative Cohort Study [MCCS], Nurses' Health Study [NHS], Prostate, Lung, Colorectal, and Ovarian Cancer Screening Trial [PLCO], and Swedish Mammography Cohort [SMC]), the association between alcohol intake and pancreatic cancer risk was evaluated after separating out former drinkers from never drinkers in the baseline non-drinkers category.

Statistical tests were two-sided with nominal level of statistical significance set to 5%. Analyses were performed using SAS version 9.4 (SAS Institute, Cary, NC, USA). No specific study protocol is available. After data harmonization, a statistical analysis plan was developed jointly between scientists at IARC and at the Harvard T.H. Chan School of Public Health (File A in S1 File). This study is reported as per the Strengthening the Reporting of Observational Studies in Epidemiology (STROBE) guideline (S1 Checklist).

## Results

The total study sample consisted of 2,494,432 participants (62% women, 70% alcohol drinkers, 47% never smokers, 64% alcohol drinkers among never smokers) in 30 studies, recruited between 1980 and 2013 with a median age of 57 years (Table 1). Within a median follow-up time of 15.6 years (10th–90th percentile range: 6.0 to 21.6 years across studies), 10,067 incident pancreatic cancer cases were diagnosed (51% women). Study participants were from North America (60%), Europe/Australia (32%) and Asia (8%) (Table 2). Participants with the highest alcohol intake were less likely to be never smokers. Median alcohol intake at recruitment among drinkers was twice as high in men (10.7 g/day) as in women (5.0 g/day) (Table A in S1 File). Cohorts from Asia (77% men) displayed the lowest percentage of alcohol drinkers (38%) and the highest median alcohol intake among drinkers (23 g/day).

We observed a statistically significant positive association between alcohol intake and pancreatic cancer risk (Fig 1). In women, HRs comparing alcohol intake of 5 to <15, 15 to <30 and ≥30 g/day to the reference category (0.1 to <5 g/day) were 0.91 (95% CI [0.84,0.99]), 1.12 (95% CI [1.00,1.25]) and 1.13 (95% CI [0.99,1.29]), respectively, with HR for a 10 g/day increment in alcohol intake ($HR_{10 g/day}$) equal to 1.03 (95% CI [1.01,1.06]). In men, HRs comparing alcohol intake of 5 to <15, 15 to <30, 30 to <60 and ≥60 g/day to the reference category were equal to 0.99 (95% CI [0.91, 1.08]), 1.02 (95% CI [0.92, 1.12]), 1.15 (95% CI [1.04, 1.28]) and 1.36 (95% CI [1.20, 1.55]), respectively, and $HR_{10 g/day}$ was equal to 1.03 (95% CI [1.02,1.04]). Among women, the HR comparing non-drinkers (<0.1 g/day) to the reference category (0.1 to <5 g/day) showed no significant association (HR = 0.97, 95% CI [0.90, 1.04]) while among men a significant positive association was observed (HR = 1.10, 95% CI [1.01, 1.20]). In sex-combined analysis, HRs comparing alcohol intake of 30 to <60 and ≥60 g/day to the reference category were equal to 1.12 (95% CI [1.03, 1.21]) and 1.32 (95% CI [1.18, 1.47]), respectively. The overall pancreatic cancer $HR_{10 g/day}$ was 1.03 (95% CI [1.02,1.04]). There was no evidence of heterogeneity by sex ($p_{sex}$ = 0.27, Fig 1) or study ($p_{study}$ = 0.40, Fig A in S1 File).

The association between alcohol intake and pancreatic cancer risk was not different by smoking status (Fig 2). The $HR_{10 g/day}$ were 1.03 (95% CI [1.01, 1.06]) in never smokers (including 3,801 cases), 1.02 (95% CI [1.00,1.04]) in former smokers (3,490 cases) and 1.03 (95% CI [1.01, 1.04]) in current smokers (2,573 cases), with p-value for heterogeneity by smoking status equal to 0.624.

There was no evidence of effect modification of the total alcohol intake and pancreatic cancer risk association by BMI, prevalent diabetes, education, and multi-vitamin use with p-values for heterogeneity equal to 0.149, 0.608, 0.284, and 0.986, respectively (Fig C in S1 File). There was significant heterogeneity by follow-up time with attenuated $HR_{10 g/day}$ estimates in the 0–2 and 5–10 years follow-up ranges ($p_{heterogeneity}$ < 0.001, Fig C in S1 File).

Positive associations of alcohol intake and pancreatic cancer risk were observed in Europe–Australia ($HR_{10 g/day}$ = 1.03, 95% CI [1.00, 1.05]) and North America ($HR_{10 g/day}$ = 1.03, 95% CI [1.02, 1.05]), while no association was observed in Asia, the region with the fewest number of alcohol drinkers in our analyses (p-value for heterogeneity by region = 0.003, Fig 3).

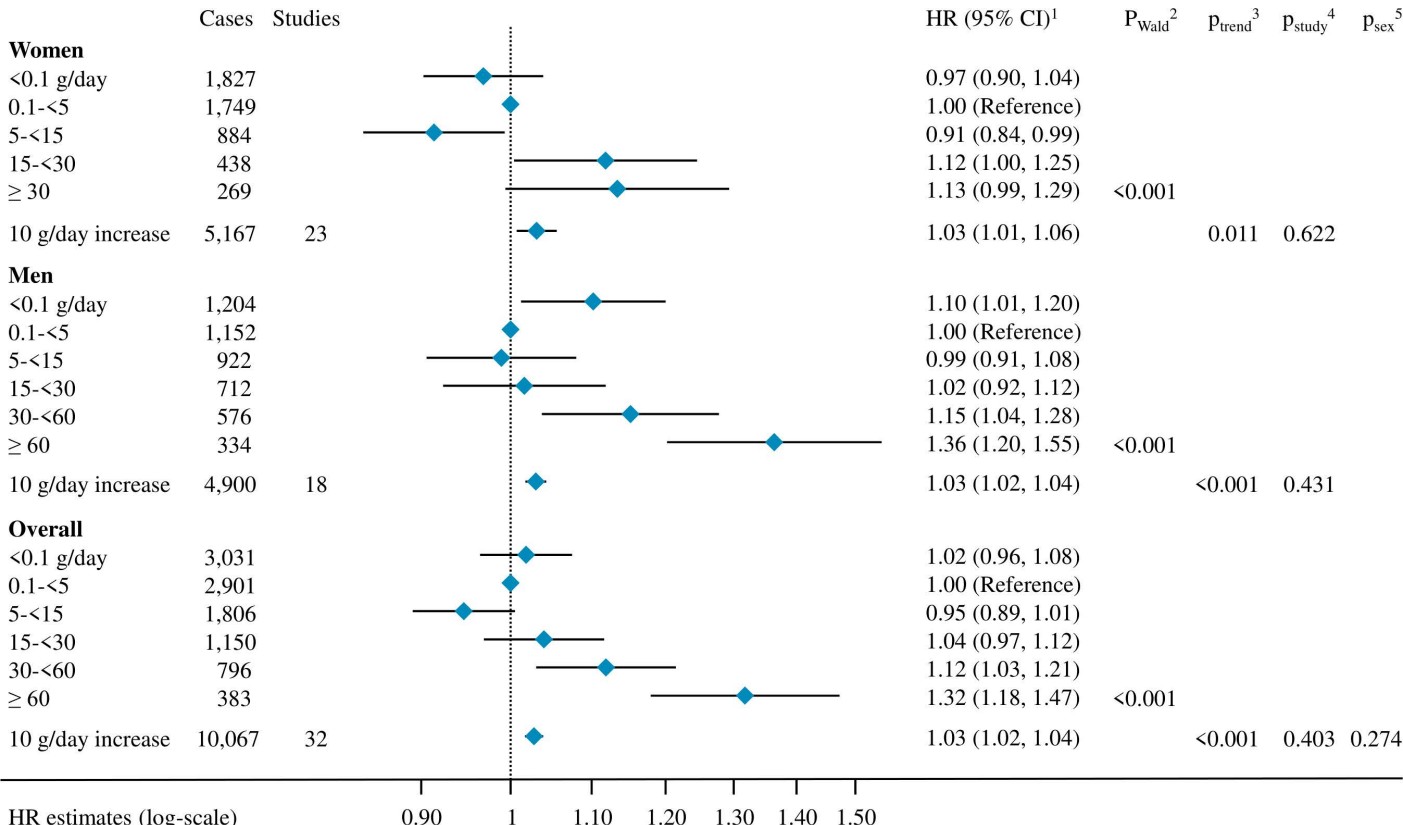

| | Cases | Studies | | HR (95% CI)[1] | $P_{Wald}$[2] | $p_{trend}$[3] | $p_{study}$[4] | $p_{sex}$[5] |
|---|---|---|---|---|---|---|---|---|
| **Women** | | | | | | | | |
| <0.1 g/day | 1,827 | | | 0.97 (0.90, 1.04) | | | | |
| 0.1-<5 | 1,749 | | | 1.00 (Reference) | | | | |
| 5-<15 | 884 | | | 0.91 (0.84, 0.99) | | | | |
| 15-<30 | 438 | | | 1.12 (1.00, 1.25) | | | | |
| ≥ 30 | 269 | | | 1.13 (0.99, 1.29) | <0.001 | | | |
| 10 g/day increase | 5,167 | 23 | | 1.03 (1.01, 1.06) | | | 0.011 | 0.622 |
| **Men** | | | | | | | | |
| <0.1 g/day | 1,204 | | | 1.10 (1.01, 1.20) | | | | |
| 0.1-<5 | 1,152 | | | 1.00 (Reference) | | | | |
| 5-<15 | 922 | | | 0.99 (0.91, 1.08) | | | | |
| 15-<30 | 712 | | | 1.02 (0.92, 1.12) | | | | |
| 30-<60 | 576 | | | 1.15 (1.04, 1.28) | | | | |
| ≥ 60 | 334 | | | 1.36 (1.20, 1.55) | <0.001 | | | |
| 10 g/day increase | 4,900 | 18 | | 1.03 (1.02, 1.04) | | | <0.001 | 0.431 |
| **Overall** | | | | | | | | |
| <0.1 g/day | 3,031 | | | 1.02 (0.96, 1.08) | | | | |
| 0.1-<5 | 2,901 | | | 1.00 (Reference) | | | | |
| 5-<15 | 1,806 | | | 0.95 (0.89, 1.01) | | | | |
| 15-<30 | 1,150 | | | 1.04 (0.97, 1.12) | | | | |
| 30-<60 | 796 | | | 1.12 (1.03, 1.21) | | | | |
| ≥ 60 | 383 | | | 1.32 (1.18, 1.47) | <0.001 | | | |
| 10 g/day increase | 10,067 | 32 | | 1.03 (1.02, 1.04) | | | <0.001 | 0.403 | 0.274 |

HR estimates (log-scale)     0.90  1  1.10  1.20  1.30  1.40  1.50

**Fig 1. Association between alcohol intake and the risk of pancreatic cancer, overall and by sex.** Abbreviations: HR: hazard ratio, CI: confidence interval; [1] Cox proportional hazard models were adjusted for smoking status, smoking duration, smoking intensity, time since smoking cessation, diabetes status, body mass index, height, education, race and ethnicity, and physical activity. Analyses in continuous were further adjusted for an indicator variable for alcohol drinking status. Models were stratified by age at baseline, year of baseline questionnaire completion, study, country (in EPIC [59]), and sex (overall models only); [2] P-value for the Wald test statistics compared with a $X^2$ distribution with degrees of freedom equal to the number of alcohol intake categories minus one, not including the category of non-drinkers (<0.1 g/day); [3] P-value for alcohol consumption modelled as a continuous variable for a 10 g/day increase, with inclusion in the model of an indicator variable expressing the alcohol drinking status; [4] Heterogeneity across studies was tested by adding interaction terms between alcohol intake modeled in continuous and each study level, then comparing the Wald test statistics for significance to a $X^2$ distribution with the number of degrees of freedom equal to the number of studies minus one, in a model including an indicator variable expressing the alcohol drinking status; [5] Heterogeneity by sex was tested by adding interaction terms between alcohol intake modelled in continuous and sex, then comparing the Wald test statistics for significance to a $X^2$ distribution with one degree of freedom, in a model including an indicator variable expressing the alcohol drinking status. The alcohol and pancreatic cancer dose–response relationship using restricted cubic splines among participants with alcohol intake lower than 100 g/day showed no departure from linearity, neither overall ($p_{nonlinearity}$ = 0.345) nor in women ($p_{nonlinearity}$ = 0.355) or men ($p_{nonlinearity}$ = 0.633) (Fig B in S1 File).

Examination of alcohol intake from different alcoholic beverages showed positive associations with pancreatic cancer risk for alcohol intake from beer and spirits/liquor with $HR_{10 g/day}$ estimates of 1.02 (95% CI [1.00, 1.04]) and 1.04 (95% CI [1.03, 1.06]), respectively, but not from wine ($HR_{10 g/day}$ = 1.00, 95% CI [0.98, 1.03], Fig 4). Associations did not differ by sex or smoking status for alcohol intake from all three beverages (Fig D in S1 File). Pancreatic cancer HR estimates for alcohol intake from wine differed by geographic region (Europe/Australia: $HR_{10 g/day}$ = 1.00, 95% CI [0.96, 1.04]; North-America: $HR_{10 g/day}$ = 1.04, 95% CI [1.00, 1.07]; Asia: $HR_{10 g/day}$ = 0.85 95% CI [0.77, 0.94]; $p_{region}$ < 0.001). Associations with alcohol intake from spirits/liquors differed by geographic region (Europe/Australia: $HR_{10 g/day}$ = 1.09, 95% CI [1.03, 1.14]; North-America: $HR_{10 g/day}$ = 1.05, 95% CI [1.03, 1.07]; Asia: $HR_{10 g/day}$ = 1.00, 95% CI [0.97, 1.04]; $p_{region}$ = 0.023).

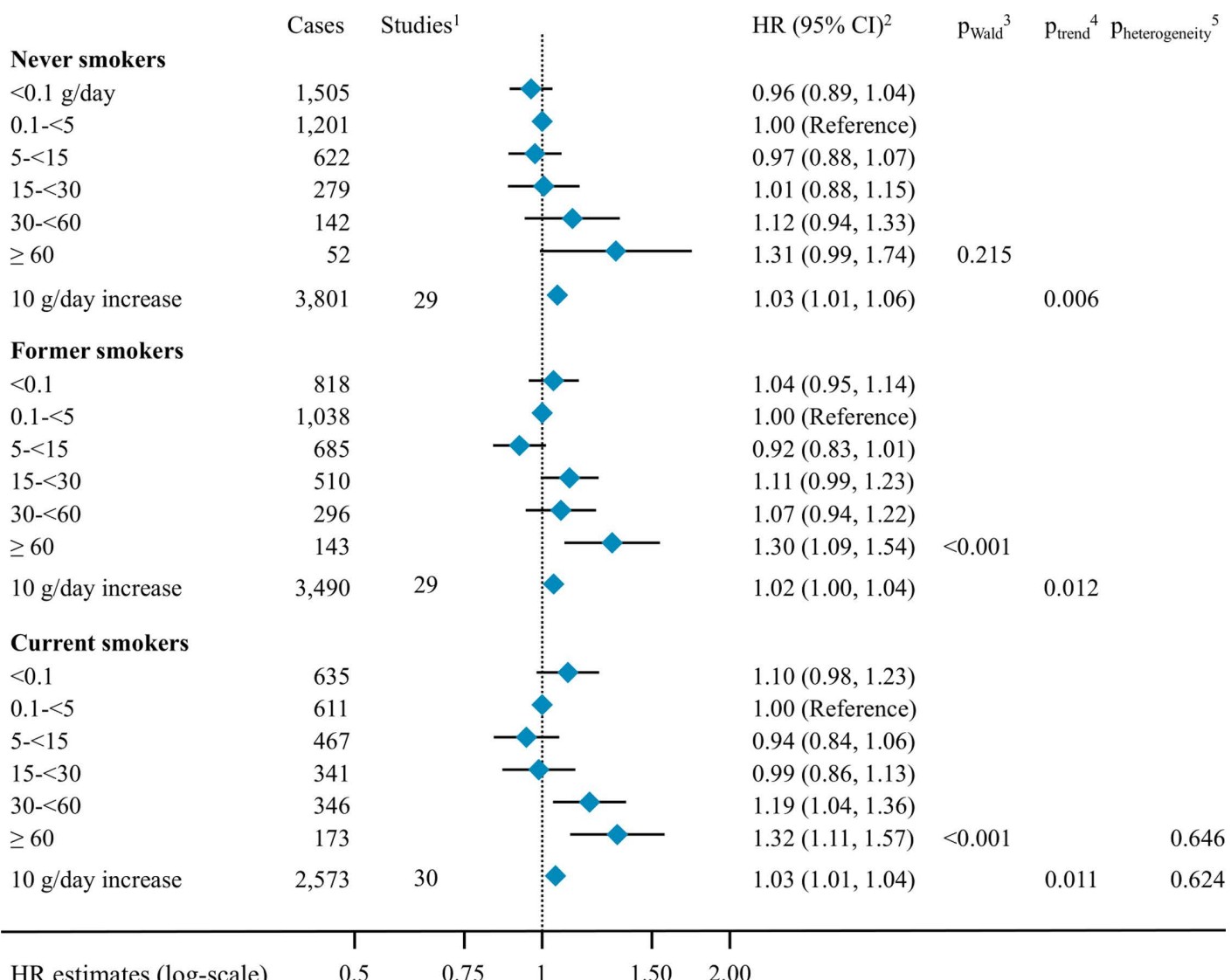

**Fig 2. Association between alcohol intake and the risk of pancreatic cancer by smoking status (never, former, current smoker).** Abbreviations: HR: hazard ratio, CI: confidence interval; [1] ATBC recruited only current smokers; [2] Cox proportional hazard models were adjusted for smoking duration, smoking intensity, time since smoking cessation, diabetes status, body mass index, height, education, race and ethnicity and physical activity. Analyses in continuous were further adjusted for an indicator variable for alcohol drinking status. Models were stratified by age at baseline, year of baseline questionnaire completion, study, country (in EPIC [59]) and sex. Models included interaction terms between alcohol intake and smoking status, keeping the 0.1 to <5 g/day category as reference, while participants without information on their smoking status were excluded; [3] P-value for the Wald test statistics compared with a $X^2$ distribution with four degrees of freedom, not including the category of non-drinkers (<0.1 g/day); [4] P-value for alcohol consumption modelled in continuous, in a model including an indicator variable expressing alcohol drinking status; [5] Heterogeneity by smoking was tested comparing the Wald test statistics for interaction between alcohol intake and smoking level to a $X^2$ distribution, with either four degrees of freedom not including the category of non-drinkers (<0.1 g/day) for analyses in categories, or one degree of freedom in a model including an indicator variable expressing alcohol drinking status for analyses in continuous.

Sensitivity analyses examining HR estimates in models with different levels of adjustment showed attenuation of $HR_{10 \text{ g/day}}$ from 1.04 to 1.03 after adjusting for smoking status versus no adjustment (model 2 versus model 1), while further adjustment for smoking duration, intensity, time since smoking cessation (model 3) and other covariates

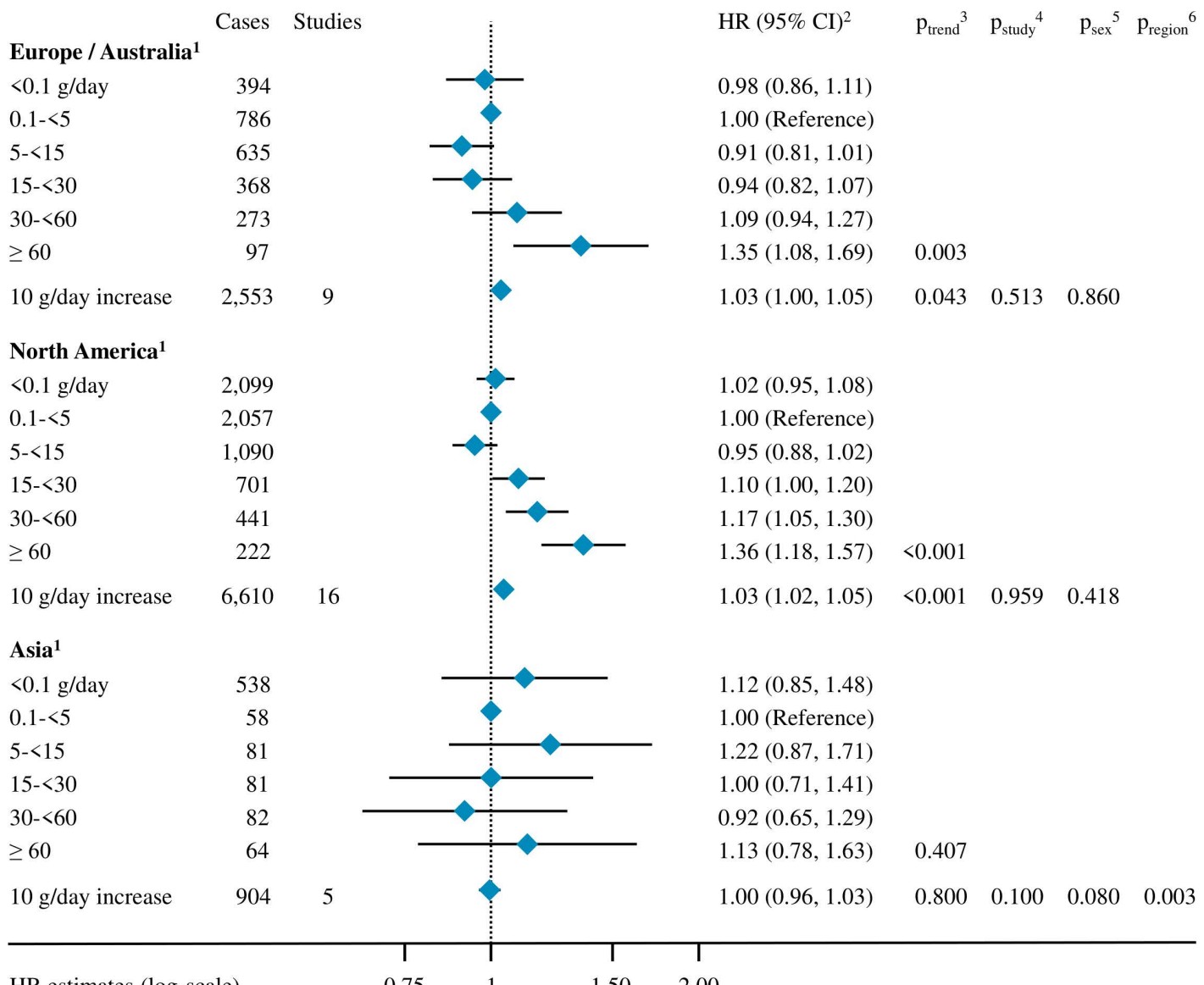

**Fig 3. Association between alcohol intake and the risk of pancreatic cancer by geographic region (Europe/Australia, North America, Asia).**
Abbreviations: HR: hazard ratio, CI: confidence interval; [1] Geographic region was coded as Europe (ATBC, COSM, GS, EPIC, MCCS, NLCS, SMC, SNMC and WLHS), North America (BCDDP, CARET, CLUEII, CNBSS, CPSII, CTS, HPFS, IWHS, MEC, NHS, NHSII, NIH-AARP, NYSC, PLCO, VITAL, and WHI) and Asia (JPHCI, JPHCII, SCHS, SCS and SMHS); [2] Cox proportional hazard models were adjusted for smoking status, smoking duration, smoking intensity, time since smoking cessation, diabetes status, body mass index, height, education, race and ethnicity, and physical activity. Analyses in continuous were further adjusted for an indicator variable for alcohol drinking status. Models were stratified by age at baseline, year of baseline questionnaire completion, study, and country (in EPIC [59]) and sex; [3] For analyses in categories, $p$-value compared the Wald test statistics with a $X^2$ distribution with degrees of freedom equal to the number of alcohol intake categories minus one, not including the category of non-drinkers (<0.1 g/day). In continuous analyses, it was the $p$-value for alcohol consumption in continuous in a model including an indicator variable expressing alcohol drinking status; [4] Heterogeneity across studies within each geographic region was tested adding interaction terms between alcohol intake modelled in continuous and each study level, then comparing the Wald test statistics for significance to a $X^2$ distribution with the number of degrees of freedom equal to the number of studies minus one, in a model including an indicator variable expressing alcohol drinking status; [5] Heterogeneity by sex within each geographic region was tested adding interaction terms between alcohol intake in continuous and sex, then comparing the Wald test statistics for significance to a $X^2$ distribution with one degree of freedom in a model including an indicator variable expressing alcohol drinking status; [6] Heterogeneity by geographic region was tested adding interaction terms between alcohol intake in continuous and geographic region, then comparing the Wald test statistics for significance to a $X^2$ distribution with two degrees of freedom in a model including an indicator variable expressing alcohol drinking status.

**Fig 4. Association between alcohol intake from different alcoholic beverages and the risk of pancreatic cancer.** Abbreviations: HR: hazard ratio, CI: confidence interval; [1] Information on type of alcoholic beverages was not available in NYSC; [2] Cox proportional hazard models were adjusted for alcohol intake from the other type of beverage than the one under evaluation, smoking status, smoking duration, smoking intensity, time since smoking cessation, diabetes status, body mass index, height, education, race and ethnicity, and physical activity. Analyses in continuous were further adjusted for an indicator variable for alcohol drinking status based on total alcohol intake. Models were stratified by age at baseline, year of baseline questionnaire completion, cohort and country (in EPIC [59]) and sex; [3] For analyses in categories, $p_{Wald}$ compared the Wald test statistics with a $X^2$ distribution with degrees of freedom equal to the number of the given type of beverage categories minus one, not including the category of non-drinkers (<0.1 g/day); [4] In continuous analyses, $p_{trend}$ was the p-value for the given type of beverage modelled as a continuous variable for a 10 g/day increase, in a model including an indicator variable expressing alcohol drinking status; [5] Heterogeneity across studies for a given type of beverage was tested adding interaction terms between the given type of beverage modeled in continuous and each study level, then comparing the Wald test statistics for significance to a $X^2$ distribution with the number of degrees of freedom equal to the number of studies minus one, in a model including an indicator variable expressing alcohol drinking status; [6] Heterogeneity by geographic region for a given type of beverage was tested adding interaction terms between the type of beverage and geographic region, then comparing the Wald test statistics for significance to a $X^2$ distribution with two degrees of freedom, in a model including an indicator variable expressing alcohol drinking status; [7] Heterogeneity by sex for each type of alcoholic beverage was tested adding interaction terms between alcohol intake and sex, then comparing the Wald test statistics for significance to a $X^2$ distribution with one degree of freedom, in a model including an indicator variable expressing alcohol drinking status.

(model 4) showed marginal further attenuation (HRs$_{10 \text{ g/day}}$ = 1.03 Fig E in S1 File). HRs were also unchanged after excluding the first 2 years of follow-up (HR$_{10 \text{ g/day}}$ = 1.03, 95% CI [1.02, 1.04]), or after restricting the case definition to histologically confirmed cases (N = 9,668 cases; HR$_{10 \text{ g/day}}$ = 1.03, 95% CI [1.02, 1.04]). In cohorts where information about past drinking status was available (n = 7), there was no association between never or former drinking and pancreatic cancer risk when compared to the category of 0.1–5 g/day in women (HR$_{never}$ = 0.93, 95% CI [0.79, 1.09]; HR$_{Former}$ = 0.98, 95% CI [0.81, 1.18]), or in men (HR$_{never}$ = 0.96, 95% CI [0.73, 1.26]; HR$_{Former}$ = 1.09, 95% CI [0,88, 1.35]) (Fig F in S1 File).

## Discussion

In a large-scale consortium of prospective cohorts, we observed a modest positive association between alcohol intake and pancreatic cancer risk, in both men and women, after controlling for a comprehensive list of potential confounding factors, including detailed information on smoking habits. Positive associations of similar magnitude were found in never, past, and current smokers.

In two previous meta-analyses that combined data from 18 and 21 cohorts (with 10 and 16 cohorts overlapping with our analysis, respectively) [10,17], a positive association with pancreatic cancer risk was reported for alcohol intake greater than 45 g/day, when compared to non-drinkers. In our previous study in DCPP based on 14 cohorts including 2,187 pancreatic cancer cases, there was a positive association with pancreatic cancer risk with alcohol intake of at least 30 g/day, as compared to non-drinkers [11]. In the current study that included more than four times the number of cases, compared to light drinkers, a significant modest positive association was observed for alcohol intakes of 30 g/day or more in men and 15 g/day or more in women; the association was stronger with alcohol intake of 60 g/day or more. Positive associations were observed in cohorts from Europe–Australia and North America, while a null association was observed in cohorts from Asia (representing 8% of the study population). No heterogeneity by geographic region was documented in previous meta-analyses [10,16,17], where few Asian cohorts were included as well [17].

Alcohol metabolism is controlled by enzymatic reactions in which alcohol dehydrogenase (ADH) converts ethanol into acetaldehyde, and aldehyde dehydrogenase (ALDH) converts acetaldehyde into acetate. The efficiency of these processes depends on variants of the *ADH* and *ALDH* genes [60,61]. Compared to Whites, Asian populations have a higher prevalence of genes encoding for the fast ADH metabolizer and slow ALDH metabolizer, which naturally leads to acetaldehyde accumulation in the bloodstream [62,63]. Carriers of these genotypes experience flushing reactions in response to alcohol ingestion and tend to drink less alcohol or abstain [15,64]. In our study, genetic data was not available. Asian populations were under-represented, with 5 cohorts out of 30 and a total of 904 incident pancreatic cancer cases included. These figures, together with the large proportion of non-drinkers, particularly among Asian women, might explain the null association observed in Asian cohorts. A larger proportion of participants in the Asian cohorts were non-drinkers at baseline (62% overall, 82% in women, 55% in men) compared to other region (15% in Europe/Australia and 33% in North America), while the median alcohol intake among male drinkers from Asian cohorts was higher than among drinkers from other regions (Table A in S1 File). These figures are consistent with previous observations from other Asian cohorts [65,66].

Several mechanisms of carcinogenesis have been suggested for alcohol intake (as ethanol), including the promotion of inflammation, microbiome dysbiosis, production of reactive oxygen species (ROS), lipid peroxidation, and DNA damage [61]. These mechanisms were suggested to cause pancreatic acinar cells injury, activate pancreatic stellate cells, and trigger pancreas fibrosis in *in vitro* models [67,68]. In a recent observational study that related alcohol intake to untargeted metabolites in EPIC and ATBC, 2-hydroxy-3-methylbutyric acid, a product of branched amino-acid metabolism correlated with alcohol intake, was positively associated with pancreatic cancer risk, suggesting the existence of a candidate molecular pathway involving fatty acyls in the alcohol related carcinogenesis of the pancreas [69]. In addition, alcohol consumption is strongly correlated with smoking habits, an established risk factor for pancreatic cancer [5]. In our study, HR

estimates were similar in never, past, and current smokers, suggesting the effect of alcohol on pancreas carcinogenesis could be independent of smoking behaviour. To our knowledge, these results are novel and shed light on inconclusive findings from previous large studies conducted in North-America, Europe and Asia [11–15].

Analyses by alcoholic beverages showed that pancreatic cancer risk was positively associated with alcohol intake from beer and spirits/liquor, while no association was observed with alcohol intake from wine. This is in line with some previous studies [9,12–14,17], but not with the previous DCPP evaluation [11] where no associations were found when alcohol intakes were evaluated separately by beverage type, although the highest category examined in those analyses was smaller (≥5 g/day), owing to the smaller number of cases. In our current analysis, risk was only notably higher for alcohol intakes from beer and from spirits/liquor of at least 20 g/day.

In this study participants drinking 0.1 to 5 g/day at baseline were chosen as the reference category throughout our evaluation, rather than using alcohol non-drinkers. Still, an unknown proportion of participants who reported low or no alcohol intake at baseline may have reduced or quit alcohol drinking before study enrolment, possibly as a result of chronic conditions like chronic pancreatitis [70–72], a strong risk factor for pancreatic cancer, which was not available in our study. If the reason for reducing alcohol consumption was a strong risk factor for pancreatic cancer (as is chronic pancreatitis), this would lead to reverse causation. To mitigate potential bias in the evaluated associations, HR estimates were evaluated excluding the first 2 years of follow-up, and findings were materially unchanged.

A major strength of this study was the size and the wide range of alcohol intake in the study population that was recruited from different geographic regions worldwide. By including more than 10,000 pancreatic cancer cases from investigations conducted in North America, Europe, Australia, and Asia, the study had greater statistical power than previous evaluations. Participating cohorts provided detailed information on pancreatic tumours, and exposure data were collected prior to cancer diagnosis. Study-specific alcohol intake, relevant covariates and information about pancreatic cancer cases were harmonized across studies to reduce potential sources of heterogeneity between studies. Models were adjusted for several potential confounders. This framework enabled a comprehensive examination of the association between alcohol intake and pancreatic cancer risk overall, as well as by sex, smoking status, education level, geographic region, and type of alcoholic beverages.

The study also had limitations. Although, pooling individual-level data enabled adjustment for a comprehensive list of confounders, we cannot rule out potential bias from unmeasured confounders. Self-reported alcohol intake is prone to systematic measurement error, as participants may under-report their alcohol intake, especially among heavy drinkers. It may result in overestimated HRs and biased associations [73], although questionnaire-based alcohol assessments showed high validity to address recall bias in many cohorts of our consortium [44–53]. Additionally, alcohol intake evaluated in this study expressed participants' average intake in grams of ethanol per day over the year preceding baseline, and did not account for alcohol intakes earlier in life, for example during early-adulthood [74]. However, in a previous study based on the EPIC cohort, baseline and lifetime alcohol intake showed similar positive associations with pancreatic cancer risk [14]. In addition, the present study did not evaluate the impact of specific drinking patterns, for example characterised by large amounts over short durations (binge drinking), due to lack of specific information. Future studies leveraging longitudinal assessments of alcohol intake from early to mid-adulthood [74] may provide insights on into the impact of alcohol drinking at different ages on pancreatic cancer risk. Finally, although data from 30 cohorts were pooled in this study, further collaborative efforts are needed to provide more comprehensive evaluations of the alcohol-PC association for specific tumor subtypes, and in geographic regions that were not included or under-represented in this study.

Findings from this large consortium of prospective studies support a modest positive association between alcohol intake and pancreatic cancer risk, irrespective of smoking status and sex. Associations were particularly evident for baseline alcohol intake of at least 15 g/day in women and 30 g/day in men. These results will inform future experts' evaluations on the epidemiological evidence of the carcinogenicity of alcohol intake.

## Patients and public involvement

Participants or the public were not involved in the design and the conduct of this study. However, these findings will have a strong translational component. They will inform general practitioners in their daily advice to the general public. More oriented dissemination activities, involving the general population and cancer patients, could be organised to emphasise the importance of reducing or quitting alcohol consumption for cancer prevention.

## Supporting information

**S1 File. Supporting information. Fig A**: Cohort-specific associations between alcohol intake, expressed for a 10 g/day increase, and the risk of pancreatic cancer. **Fig B**: Pancreatic cancer hazard ratios (solid line) and corresponding 95% confidence interval (dashed line) as a function of alcohol intake (ranging from 0 to 100 g/day). **Fig C**: Heterogeneity in the alcohol-pancreatic cancer association, for alcohol intake expressed for a 10 g/day increase, by body mass index, diabetes status, education, follow-up time and multivitamin use. **Fig D**: Association between alcohol intake and the risk of pancreatic cancer by type of alcoholic beverage and by geographic region, sex, education, and smoking status. **Fig E**: Association between alcohol intake and the risk of pancreatic cancer using different levels of adjustment for smoking habits and pancreatic cancer risk factors. **Fig F**: Association between alcohol intake and the risk of pancreatic cancer among studies without (left) and with (right) information on past drinking (COSM, EPIC, HPFS, MCCS, NHS, PLCO, SMC). **Table A**: Study, region, and sex-specific alcoholic beverage intake among drinkers. **Table B**: Study institutional review board and approval number. **File A**: Statistical analysis plan for evaluation of the association between alcohol intake and pancreatic cancer risk within the Pooling Project on Alcohol and Cancer.
(DOCX)

**S1 Checklist. STROBE Statement.** Manuscript checklist according to the Strengthening the Reporting of Observational Studies in Epidemiology (STROBE) guideline for cohort studies.
(DOCX)

## Acknowledgments

**For the Consortium:** We thank the participants and investigators of the 30 cohorts involved in this project, the NCI Cohort Consortium, Ms Carine Biessy for support in data analysis and quality control, and Ms Amelia Zhang-Gross for administrative support without whom this work would have never been possible.

  **Cancer Prevention Study-II Nutrition Cohort (CPS II):** The authors express sincere appreciation to all Cancer Prevention Study-II participants, and to each member of the study and biospecimen management group. The authors would like to acknowledge the contribution to this study from central cancer registries supported through the Centers for Disease Control and Prevention's National Program of Cancer Registries and cancer registries supported by the National Cancer Institute's Surveillance Epidemiology and End Results Program.

  **Health Professionals Follow-Up Study (HPFS):** The authors would like to acknowledge the contribution to this study from central cancer registries supported through the Centers for Disease Control and Prevention's National Program of Cancer Registries (NPCR) and/or the National Cancer Institute's Surveillance, Epidemiology, and End Results (SEER) Program. We are grateful to the study participants and staff of the HPFS cohort.

  **Nurses' Health Study (NHS):** The authors would like to acknowledge the contribution to this study from central cancer registries supported through the Centers for Disease Control and Prevention's National Program of Cancer Registries (NPCR) and/or the National Cancer Institute's Surveillance, Epidemiology, and End Results (SEER) Program. The content is solely the responsibility of the authors and does not necessarily represent the official views of the National Institutes of Health.

**Nurses' Health Study II (NHS II):** The authors would like to acknowledge the contribution to this study from central cancer registries supported through the Centers for Disease Control and Prevention's National Program of Cancer Registries (NPCR) and/or the National Cancer Institute's Surveillance, Epidemiology, and End Results (SEER) Program.

**NIH-AARP Diet and Health Study (NIH-AARP):** We are indebted to the participants in the NIH-AARP Diet and Health Study for their cooperation. We also thank Sigurd Hermansen and Kerry Grace Morrissey from Westat for NIH-AARP study outcomes ascertainment and management and Leslie Carroll at Information Management Services for data support and analysis.

**Prostate, Lung, Colorectal, and Ovarian Cancer Screening Trial (PLCO):** The authors thank the NCI for access to NCI's data collected by the Prostate, Lung, Colorectal and Ovarian (PLCO) Cancer Screening Trial, as well as the NCI study management team, the screening center investigators, and staff at Information Management Services, and Westat, Most importantly, we thank the study participants for their contributions that made this study possible.

**Singapore Chinese Health Study (SCHS):** We thank Siew-Hong Low of the National University of Singapore for supervising the field work of the Singapore Chinese Health Study. We also thank the Singapore Cancer Registry for the identification of incident cancer cases among participants of the Singapore Chinese Health Study. We have no conflict of interests to declare.

## Author contributions

**Conceptualization:** Sabine Naudin, Molin Wang, Jeanine Genkinger, Tao Hou, Shiaw-Shyuan S. Yaun, Paul Brennan, Stephanie A. Smith-Warner, Pietro Ferrari.

**Data curation:** Sabine Naudin, Molin Wang, Jeanine Genkinger, Hans-Olov Adami, Demetrius Albanes, Ana Babic, Matt Barnett, David Bogumil, Hui Cai, Chu Chen, A. Heather Eliassen, Jo L. Freudenheim, Gretchen Gierach, Edward L. Giovanucci, Marc J. Gunter, Niclas Håkansson, Mayo Hirabayashi, Tao Hou, Brian Z. Huang, Wen-Yi Huang, Harindra Jayasekara, Michael E. Jones, Verena A. Katzke, Woon-Puay Koh, James V. Lacey, Jr., Ylva Trolle Langerros, Susanna C. Larsson, Linda M. Liao, Kenneth Lo, Errika Loftfield, Robert J. MacInnis, Satu Männistö, Marjorie L. McCullough, Anthony Miller, Roger L. Milne, Steven C. Moore, Lorelei A. Mucci, Marian L. Neuhouser, Alpa V. Patel, Elizabeth A. Platz, Anna Prizment, Kim Robien, Thomas E. Rohan, Carlotta Sacerdote, Sven Sandin, Norie Sawada, Minouk Shoemaker, Xiao-Ou Shu, Rashmi Sinha, Linda Snetselaar, Meir J. Stampfer, Rachael Stolzenberg-Solomon, Cynthia A. Thomson, Anne Tjønneland, Caroline Y. Um, Piet A. van den Brandt, Kala Visvanathan, Sophia S. Wang, Renwei Wang, Elisabete Weiderpass, Stephanie J. Weinstein, Emily White, Walter Willett, Alicja Wolk, Brian M. Wolpin, Shiaw-Shyuan S. Yaun, Chen Yaun, Jian-Min Yuan, Wei Zheng, Paul Brennan, Stephanie A. Smith-Warner, Pietro Ferrari.

**Formal analysis:** Sabine Naudin, Molin Wang, Niki Dimou, Elmira Ebrahimi, Tao Hou, Shiaw-Shyuan S. Yaun, Stephanie A. Smith-Warner, Pietro Ferrari.

**Funding acquisition:** Paul Brennan, Stephanie A. Smith-Warner, Pietro Ferrari.

**Investigation:** Sabine Naudin, Molin Wang, Niki Dimou, Elmira Ebrahimi, Jeanine Genkinger, Hans-Olov Adami, Demetrius Albanes, Ana Babic, Matt Barnett, David Bogumil, Hui Cai, Chu Chen, A. Heather Eliassen, Jo L. Freudenheim, Gretchen Gierach, Edward L. Giovanucci, Marc J. Gunter, Niclas Håkansson, Mayo Hirabayashi, Tao Hou, Brian Z. Huang, Wen-Yi Huang, Harindra Jayasekara, Michael E. Jones, Verena A. Katzke, Woon-Puay Koh, James V. Lacey, Jr., Ylva Trolle Langerros, Susanna C. Larsson, Linda M. Liao, Kenneth Lo, Errika Loftfield, Robert J. MacInnis, Satu Männistö, Marjorie L. McCullough, Anthony Miller, Roger L. Milne, Steven C. Moore, Lorelei A. Mucci, Marian L. Neuhouser, Alpa V. Patel, Elizabeth A. Platz, Anna Prizment, Kim Robien, Thomas E. Rohan, Carlotta Sacerdote, Sven Sandin, Norie Sawada, Minouk Shoemaker, Xiao-Ou Shu, Rashmi Sinha, Linda Snetselaar, Meir J. Stampfer, Rachael Stolzenberg-Solomon, Cynthia A. Thomson, Anne Tjønneland, Caroline Y. Um, Piet A. van den

Brandt, Kala Visvanathan, Sophia S. Wang, Renwei Wang, Elisabete Weiderpass, Stephanie J. Weinstein, Emily White, Walter Willett, Alicja Wolk, Brian M. Wolpin, Shiaw-Shyuan S. Yaun, Chen Yaun, Jian-Min Yuan, Wei Zheng, Paul Brennan, Stephanie A. Smith-Warner, Pietro Ferrari.

**Methodology:** Sabine Naudin, Molin Wang, Niki Dimou, Elmira Ebrahimi, Jeanine Genkinger, Tao Hou, Shiaw-Shyuan S. Yaun, Stephanie A. Smith-Warner, Pietro Ferrari.

**Project administration:** Sabine Naudin, Stephanie A. Smith-Warner, Pietro Ferrari.

**Resources:** Stephanie A. Smith-Warner, Pietro Ferrari.

**Software:** Sabine Naudin, Molin Wang, Tao Hou, Shiaw-Shyuan S. Yaun, Stephanie A. Smith-Warner, Pietro Ferrari.

**Supervision:** Paul Brennan, Stephanie A. Smith-Warner, Pietro Ferrari.

**Validation:** Sabine Naudin, Molin Wang, Niki Dimou, Elmira Ebrahimi, Tao Hou, Shiaw-Shyuan S. Yaun, Paul Brennan, Stephanie A. Smith-Warner, Pietro Ferrari.

**Visualization:** Sabine Naudin, Pietro Ferrari.

**Writing – original draft:** Sabine Naudin, Paul Brennan, Stephanie A. Smith-Warner, Pietro Ferrari.

**Writing – review & editing:** Sabine Naudin, Molin Wang, Niki Dimou, Elmira Ebrahimi, Jeanine Genkinger, Hans-Olov Adami, Demetrius Albanes, Ana Babic, Matt Barnett, David Bogumil, Hui Cai, Chu Chen, A. Heather Eliassen, Jo L. Freudenheim, Gretchen Gierach, Edward L. Giovanucci, Marc J. Gunter, Niclas Håkansson, Mayo Hirabayashi, Brian Z. Huang, Wen-Yi Huang, Harindra Jayasekara, Michael E. Jones, Verena A. Katzke, Woon-Puay Koh, James V. Lacey, Jr., Ylva Trolle Langerros, Susanna C. Larsson, Linda M. Liao, Kenneth Lo, Errika Loftfield, Robert J. MacInnis, Satu Männistö, Marjorie L. McCullough, Anthony Miller, Roger L. Milne, Steven C. Moore, Lorelei A. Mucci, Marian L. Neuhouser, Alpa V. Patel, Elizabeth A. Platz, Anna Prizment, Kim Robien, Thomas E. Rohan, Carlotta Sacerdote, Sven Sandin, Norie Sawada, Minouk Shoemaker, Xiao-Ou Shu, Rashmi Sinha, Linda Snetselaar, Meir J. Stampfer, Rachael Stolzenberg-Solomon, Cynthia A. Thomson, Anne Tjønneland, Caroline Y. Um, Piet A. van den Brandt, Kala Visvanathan, Sophia S. Wang, Renwei Wang, Elisabete Weiderpass, Stephanie J. Weinstein, Emily White, Walter Willett, Alicja Wolk, Brian M. Wolpin, Chen Yaun, Jian-Min Yuan, Wei Zheng, Paul Brennan, Stephanie A. Smith-Warner, Pietro Ferrari.

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
