## [Editor Report · Decision Letter 0]

19 Jul 2024

Dear Dr Ferrari, 

Thank you for submitting your manuscript entitled "Alcohol intake and pancreatic cancer risk: an analysis from 30 prospective studies" for consideration by PLOS Medicine.

Your manuscript has now been evaluated by the PLOS Medicine editorial staff and I am writing to let you know that we would like to send your submission out for external peer review.

Please re-submit your manuscript within two working days, i.e. by Jul 23 2024.

Feel free to email me at atosun@plos.org or us at plosmedicine@plos.org if you have any queries relating to your submission.

Kind regards,

Alexandra Tosun, PhD

Associate Editor

PLOS Medicine

---

## [Decision Letter · Decision Letter 1]

20 Sep 2024

Dear Dr Ferrari,

Many thanks for submitting your manuscript "Alcohol intake and pancreatic cancer risk: an analysis from 30 prospective studies" (PMEDICINE-D-24-02276R1) to PLOS Medicine. The paper has been reviewed by subject experts and a statistician; their comments are included below and can also be accessed here: [LINK]

As you will see, the reviewers found the study to be well conducted, but raised several points for clarification. After discussing the paper with the editorial team and an academic editor with relevant expertise, I'm pleased to invite you to revise the paper in response to the reviewers' comments. We plan to send the revised paper to some or all of the original reviewers, and we cannot provide any guarantees at this stage regarding publication.

We ask that you submit your revision by Oct 11 2024. However, if this deadline is not feasible, please contact me by email, and we can discuss a suitable alternative.

Don't hesitate to contact me directly with any questions (atosun@plos.org). 

Best regards, 

Alexandra 

Alexandra Tosun, PhD 

Associate Editor

PLOS Medicine

atosun@plos.org

Comments from the academic editor:

This paper examines the association between alcohol consumption and the risk of pancreatic cancer, a topic of significant global importance. The methodology, including both the study design and analysis, is exceptional, and the results are both interesting and impactful. While previous systematic reviews and meta-analyses on this topic have been published, the strength of the current study lies in its pooling of individual-level data from 31 cohort studies. This approach allows for more sophisticated analyses and provides sufficient power to examine the effects of alcohol consumption at different dose levels, with appropriate control for confounders and the ability to explore effect modification. The inclusion of international cohorts enhances the generalizability of the findings on a global scale. The key findings are of high clinical relevance and could have implications for public policy.

A few comments for the authors:

The analysis explores heterogeneity across the cohort studies by using interaction terms between alcohol intake (modeled continuously) and each study. The results indicate that there was no heterogeneity between the cohorts. However, while heterogeneity was not observed, does this eliminate the need to address clustering by cohort? Alternative approaches to account for clustering should be considered, such as hierarchical models or generalized estimating equations.

Additionally, the methodology for addressing heterogeneity, particularly heterogeneity by study, is not clearly explained in the statistical analysis section. Although the figure legend describes the approach well, I recommend including more detail in the methods section regarding how heterogeneity was handled.

Lastly, the authors mention that data harmonization was performed across the different cohorts, but they do not explain how this was achieved, given that the cohorts likely collected data differently. Further elaboration on the harmonization process would be helpful. 

Comments from the reviewers: 

Reviewer #1: This study investigates the association between alcohol drinking and pancreatic cancer risk through a pooled-analyses of 30 prospective studies, allowing the assessment of the possible modifying effect of various factors. The study is very well conducted and presented. I would only modify the main message in the abstract and conclusion, indicating that the study supports an association for high alcohol consumption.

Please correct "95%CI" in "95% CI" throughout all the Manuscript.

Abstract

The sentence "Alcohol intake was positively associated with pancreatic cancer risk, with HR5-to-<15g/day, ….." can be modified into "Alcohol intake above 30 g/day was positively associated with pancreatic cancer risk, with HR30-to-<60g/day and HR≥60g/day equal to 01.12 (95%CI:1.03,1.21) and 1.32 (95%CI: 1.18,1.47), respectively, compared to intake of 0.1-<5g/day.". 

The sentence "specifically in women (HR: 1.03; 95%CI:1.01,1.06; pvalue<0.01) and never smokers (HR: 1.03; 95%CI:1.01,1.06; pvalue<0.01) with no evidence of heterogeneity by sex (pheterogeneity=0.27) or smoking status (pheterogeneity=0.62)" can be modified as follows "No evidence of heterogeneity was observed by sex (p heterogeneity=0.27) or smoking status (p heterogeneity=0.62)".

The conclusion should be modified as follows "Findings from this large-scale pooled analysis support a modest positive association between elevated alcohol intake (at least 15g/day in women and 30g/day in men) and pancreatic cancer risk, irrespective of sex and smoking status".

Introduction

Authors may also quote this pooled analysis of case-control studies "Lucenteforte E et al. Alcohol consumption and pancreatic cancer: a pooled analysis in the International Pancreatic Cancer Case-Control Consortium (PanC4). Ann Oncol. 2012 Feb;23(2):374-82. doi: 10.1093/annonc/mdr120"

Discussion

Please revise the first sentence of the Discussion and the Conclusion following the previous suggestion.

Tables 

In the Tables, please delete the "%" sign, since this is already indicated in the first column.

Reviewer #2: This is a well-conducted study on the association between alcohol intake and pancreatic cancer risk based on an analysis from 30 prospective studies. The study design, datasets, statistical methods and analyses, and presentation (tables and figures) and interpretation of the results are mostly adequate. However, there are still a few issues needing attention.

1. Justification of inclusion of 30 studies. As we know, the gold standard of pooling evidence is a systematic review and meta-analysis through systematic search of literatures. But this study is not. As the studies included didn't go through systematic search, how its results compared to a meta-analysis? pros and cons? Are there any more recent meta-analysis on this topic?

2. As shown in the limitations in page 23, it says "exposure assessments at baseline generally relates to participants' alcohol intake during mid-adulthood, and do not account for potential within-person change in alcohol intake from adolescence to later in life, nor for changes that may occur after baseline". This is crucially important. We know people's drinking pattern changes from adolescence to mid-life and then later life, but so far this study only captured a static mid-life one. As the the included studies varies in starting times and also follow-up times, there are substantial uncertainties in un-captured drinking patterns. Besides, from GBD studies we know people are drinking less and less over the past 3 decades. However, all these trends in drinking were not captured in the study. Therefore, the robustness and reliability of findings are subject to scrutiny.

3. Competing risk. As the endpoint of the analysis is pancreatic cancer incidence, death poses a competing risk in the analysis, which has not beed addressed in the study.

Reviewer #3: This study addresses an important and timely research question - associations between alcohol consumption and pancreatic cancer where evidence was considered as limited or inconclusive by international expert panels hence updated evidence is needed. This study, with larger sample size, availability of more diverse cohorts, and greater power for subgroup analyses than previous studies, provides updated evidence to help clarify the observational association of alcohol drinking with pancreatic cancer, especially by sex and smoking status. I have several comments below for potential further improvements of the paper.

* Lines 109-110, "specifically in women…": The dose-response association seems much clearer in men than women (Figure 1), and HR per 10g/day was significant in both sexes. Is there any reason to particularly report women findings in the abstract?

* Lines 298-300, 305-306: An alternative way to quantify the effect of continuous alcohol intake would be to analyse alcohol intake (as continuous variable of g/day) among drinkers only. Have the authors considered this method, over performing the analyses in all participants while adjusting for binary alcohol status as described in the paper?

* Lines 314-326, following on above comment: are these done among drinkers only, or in all participants but adjusting for binary alcohol status variable?

* Lines 454-456: though p for heterogeneity is significant for wine by education level, the HRs themselves are non-significant. I'd suggest to be cautious and avoid over-interpretation of this heterogeneity results.

* Lines 518-520, Supplementary table 1. The median alcohol intake among drinkers is much higher in Asia than Europe-Australia/North America. The extent of gap is quite surprising. Are the any potential explanations or insights for this gap that could perhaps be discussed in the discussion and support the data?

* Null associations in Asian studies: while lightly touched upon (e.g. line 507 that Asian cohorts only represented 8% of the study population), the point that Asian studies are underrepresented and uncertainty of the null associations (e.g. potentially due to lack of power?) could perhaps be discussed a little more, or discussed as suggestion for future research/limitation.

* It would also be good to have a short discussion to pinpoint suggested direction of future studies based on the study findings (also see above comment).

* In the Abstract, the conclusion (line 122-123) mentioned that the heterogeneity by regions and beverage types might be due to different drinking habits (and hence further investigation needed), but this idea seemed not presented/discussed in the main text (particularly discussion). It would perhaps be good to introduce or, if appropriate, elaborate this a little considering also other potential explanations (e.g. power/availability of relevant studies for inclusion, differential drinking habits, other confounding factors) in the main text.

---

* Please upload any figures associated with your paper as individual TIF or EPS files with 300dpi resolution at resubmission; please read our figure guidelines for more information on our requirements: http://journals.plos.org/plosmedicine/s/figures. While revising your submission, please upload your figure files to the PACE digital diagnostic tool, https://pacev2.apexcovantage.com/. PACE helps ensure that figures meet PLOS requirements. To use PACE, you must first register as a user. Then, login and navigate to the UPLOAD tab, where you will find detailed instructions on how to use the tool. If you encounter any issues or have any questions when using PACE, please email us at PLOSMedicine@plos.org.

* FINANCIAL DISCLOSURES: The funding statement should include: specific grant numbers, initials of authors who received each award, URLs to sponsors’ websites. Also, please state whether any sponsors or funders (other than the named authors) played any role in study design, data collection and analysis, the decision to publish, or preparation of the manuscript. If they had no role in the research, include this sentence: “The funders had no role in study design, data collection and analysis, decision to publish, or preparation of the manuscript.”

* COMPETING INTEREST: All authors must declare their relevant competing interests per the PLOS policy, which can be seen here: https://journals.plos.org/plosmedicine/s/competing-interests

For authors with ties to industry, please indicate whether any of the interests has a financial stake in the results of the current study."

Please add this statement to the manuscript's Competing Interests: "WZ and SCM are Academic Editors on PLOS Medicine's editorial board."

* DATA AVAILABILITY: The Data Availability Statement (DAS) requires revision. For each data source used in your study: 

FIGURES AND TABLES

SUPPLEMENTARY MATERIAL

REFERENCES

* Where website addresses are cited, please include the complete URL and specify the date of access (e.g. [accessed: 12/06/2024]).

STUDY TYPE-SPECIFIC REQUESTS

* Abstract: Please include the study design, population and setting, number of participants, years during which the study took place (enrollment and follow up), length of follow up, and main outcome measures.

* Please ensure that the study is reported according to the STROBE (or appropriate STOBE extension) guideline (available from: https://www.equator-network.org/reporting-guidelines/strobe) and include the completed STROBE (or STROBE extension) checklist as Supporting Information. Please add the following statement, or similar, to the Methods: "This study is reported as per the Strengthening the Reporting of Observational Studies in Epidemiology (STROBE) guideline (S1 Checklist)." When completing the checklist, please use section and paragraph numbers, rather than page numbers. 

* For all observational studies, in the manuscript text, please indicate: (1) the specific hypotheses you intended to test, (2) the analytical methods by which you planned to test them, (3) the analyses you actually performed, and (4) when reported analyses differ from those that were planned, transparent explanations for differences that affect the reliability of the study's results. If a reported analysis was performed based on an interesting but unanticipated pattern in the data, please be clear that the analysis was data driven. 

* Please state in the Methods section whether the study had a prospective protocol or analysis plan. If a prospective analysis plan (from your funding proposal, IRB or other ethics committee submission, study protocol, or other planning document written before analyzing the data) was used in designing the study, please include the relevant document(s) with your revised manuscript as a Supporting Information file to be published alongside your study and cite it in the Methods section. A legend for this file should be included at the end of your manuscript. If no such document exists, please make sure that the Methods section transparently describes when analyses were planned, and when/why any data-driven changes to analyses took place. Changes in the analysis, including those made in response to peer review comments, should be identified as such in the Methods section of the paper, with rationale.

---

## [Decision Letter · Decision Letter 2]

31 Jan 2025

Dear Dr. Ferrari,

Thank you very much for re-submitting your manuscript "Alcohol intake and pancreatic cancer risk: an analysis from 30 prospective studies" (PMEDICINE-D-24-02276R2) for review by PLOS Medicine.

Thank you for your detailed response to the editors' and reviewers' comments. I have discussed the paper with my colleagues and the academic editor, and it has also been seen again by two of the original reviewers. The changes made to the paper were satisfactory to the reviewers. As such, we intend to accept the paper for publication, pending your attention to the reviewers' and editors' comments below in a further revision. When submitting your revised paper, please once again include a detailed point-by-point response to the editorial comments.

[LINK]

In revising the manuscript for further consideration here, please ensure you address the specific points made by each reviewer and the editors. In your rebuttal letter you should indicate your response to the reviewers' and editors' comments and the changes you have made in the manuscript. Please submit a clean version of the paper as the main article file. A version with changes marked must also be uploaded as a marked up manuscript file. Please also check the guidelines for revised papers at http://journals.plos.org/plosmedicine/s/revising-your-manuscript for any that apply to your paper.

We ask that you submit your revision within 1 week (Feb 07 2025). However, if this deadline is not feasible, please contact me by email, and we can discuss a suitable alternative.

Please do not hesitate to contact me directly with any questions (atosun@plos.org). If you reply directly to this message, please be sure to 'Reply All' so your message comes directly to my inbox.

We look forward to receiving the revised manuscript. 

Sincerely,

Alexandra Tosun, PhD

Associate Editor 

PLOS Medicine

plosmedicine.org

Comments from Reviewers:

Reviewer #2: Thanks authors for their effort to improve the manuscript. I am satisfied with the response and revision. No further issues needing attention.

Reviewer #3: The authors have addressed my previous comments satisfactorily. I have no further comments or concerns at this stage.

[LINK]

Requests from Editors:

GENERAL

The terms gender and sex are not interchangeable (as discussed in https://www.who.int/health-topics/gender#tab=tab_1 ); please use the appropriate term and revise accordingly throughout the manuscript. Please note that you currently use both.

FINANCIAL DISCLOSURE

Please remove the Acknowledgments section from the Financial Disclosure section of the online submission form. You may include the Acknowledgments at the end of the main manuscript, which will appear as a separate section in the final publication.

DATA AVAILABILITY

We currently feel that it is not 100% clear how others may request data. Please note that we require you to provide an appropriate contact (web or email address) for requests (this cannot be a study author). Please clarify whether the contact at the Harvard T.H. Chan School of Public Health will be the only point of contact for data requests for any of the study cohorts.

TITLE

If you agree, we suggest changing the title to "Alcohol intake and pancreatic cancer 1 risk: an analysis from 30 prospective studies across Asia, Australia, Europe, and North America" to reflect the broad scope of your study.

ETHICS

We ask that you provide a list of institutional review boards and corresponding approval numbers. This can be included in the Supplementary Material and referenced in the main text.

ABSTRACT

1) ll.98/99: PLOS Medicine prefers the use of patient-centered language. Please revise throughout the manuscript.

2) l.99: “alcohol drinkers” – we suggest adding ‘(alcohol intake ≥ 0.1g/day)’ here.

3) l.105ff: Throughout, we suggest reporting statistical information as follows to improve clarity for the reader "1.12 (95% CI [1.03,1.21])".

4) In the last sentence of the Abstract Methods and Findings section, please describe the main limitation(s) of the study's methodology.

5) Please ensure that all numbers presented in the abstract are present and identical to numbers presented in the main manuscript text.

6) Please include the important dependent variables that are adjusted for in the analyses.

AUTHOR SUMMARY

ll.128-129: Since the Author Summary serves as a non-technical summary of your research in order to make the results accessible to a wide audience, including scientists and non-scientists, we suggest adding a brief note on what 30 g of alcohol per day is equivalent to, e.g. 10 oz or 300 ml of table wine.

INTRODUCTION

1) Please change the heading from ‘Background’ to ‘Introduction’.

2) Please cite the reference numbers in square brackets. Please revise throughout.

3) l.224: Please define ‘US’ at first use.

METHODS AND RESULTS

1) Table 1: Please define ‘SD’ (l.266). Also, when reporting percentiles, please separate upper and lower bounds with commas instead of hyphens as the latter can be confused with reporting of negative values. Please revise throughout the manuscript.

2) l.323ff: We think it would be easier for the reader if you provided not only the percentage, but also the numerator and denominator. Please revise throughout where necessary.

3) l. 326: Please define ‘BMI’ at first use.

4) Table 2 (and where applicable): Please present numerators and denominators for percentages. When presenting the median, please also provide 10th–90th percentiles. 

5) Figure 1/2/3/4: Please define BMI, HR and CI. Please add a reference to the EPIC study.

6) Figure 1/2/4 and Table 1: We think that for each category, the included studies should be listed, e.g. in a footnote, as was done for Figure 3, and the relevant references should be provided.

7) Figure 2/4: Could you explain why you decided to include the number of studies in Figure 1 and 3 but not in Figure 2 and 4?

8) Figure 3: We suggest adding the corresponding reference numbers for the studies listed in the figure description.

9) Please include a brief explanation of why Australia was grouped with Europe, as we believe you did not include an explanation for this in the Methods section.

DISCUSSION

1) l.573: Please temper claims of primacy of results by stating, "to our knowledge" or something similar.

2) Please remove any subheadings. The conclusion should be a continuous part of the discussion.

REFERENCES

Where website addresses are cited, please use the word ‘accessed’ to specify the date of access (e.g. [accessed: 12/06/2024]).

SUPPLEMENTARY MATERIAL

1) Thank you for providing the STROBE checklist. Please replace the page numbers with paragraph numbers per section (e.g. "Methods, paragraph 1"), since the page numbers of the final published paper may be different from the page numbers in the current manuscript.

2) In the published article, supporting information files are accessed only through a hyperlink attached to the captions. For this reason, you must list captions at the end of your manuscript file. You may include a caption within the supporting information file itself, as long as that caption is also provided in the manuscript file. Do not submit a separate caption file.

When SI files are contained with a single file:

Please label the file as ‘S1 Supporting Information’.

Please apply alphabetical labelling to each table and figure contained within the S1 file. For example, ‘Fig A’ to ‘Fig Z’ and ‘Table A’ to ‘Table Z’.

Plain text does not need to be labelled and can just be given a title as necessary. For example, ‘Statistical Analysis Plan’.

Please cite tables/figures as ‘Fig A in S1 Supporting Information’ and/or ‘Table A in S1 Supporting Information’, for example.

Please cite plain text as, ‘Statistical Analysis Plan in S1 Supporting Information’, for example.

When SI files are uploaded as separate files:

Please label tables as ‘S1 Table’ (so on) and figures as ‘S1 Fig’ (and so on).

Any additional documents (protocols/analysis plans etc.) can be labelled as ‘S1 Protocol’, for example. Please cite items as exactly as labelled.

General Editorial Requests

---

## [Editor Report · Decision Letter 3]

14 Mar 2025

Dear Dr. Ferrari,

Thank you very much for re-submitting your manuscript "Alcohol intake and pancreatic cancer risk: an analysis from 30 prospective studies across Asia, Australia, Europe, and North America" (PMEDICINE-D-24-02276R3) for review by PLOS Medicine.

There are a few minor editorial issues that need to be addressed before we can accept the manuscript for publication; these are outlined at the end of this email. Please revise the paper accordingly, and submit the final revision within 1 week (Mar 21).

Please ensure you address the specific points made by the editors. In your rebuttal letter you should indicate your response to the editors' comments and the changes you have made in the manuscript. Please submit a clean version of the paper as the main article file. A version with changes marked must also be uploaded as a marked up manuscript file. Please also check the guidelines for revised papers at http://journals.plos.org/plosmedicine/s/revising-your-manuscript for any that apply to your paper. 

A reminder that when your manuscript is accepted, an uncorrected proof of your manuscript will be published online ahead of the final version, unless you've already opted out via the online submission form. If, for any reason, you do not want an earlier version of your manuscript published online or are unsure if you have already indicated as such, please let the journal staff know immediately at plosmedicine@plos.org.

If you have any questions in the meantime, please contact me directly at atosun@plos.org.

We look forward to receiving the revised manuscript.

Sincerely,

Alexandra Tosun, PhD

Associate Editor

PLOS Medicine

Requests from Editors:

1) Ethics statement: Please note that we cannot accept the manuscript until you have provided a complete list of the institutional review boards together with the relevant approval numbers. Again, this can be added in the Supplementary Material and referenced in the main text.

2) Author Summary: We feel that the main limitation could be more clearly stated in the last bullet point. Please revise.

3) Table 2: Please clarify how you calculated the % values for the number of PC cases. For most of the characteristics it seems that the denominator is the total number of male or female participants in the respective group. However, this is not the case for number of PC cases. Please add a short explanation below the table that for categorical variables the denominator for each group of alcohol intake is the number of participants in the corresponding group and explain the exception for number of PC cases. Please also define "PC" below the table.

4) Funding: Please include a short statement in the Financial Disclosure Statement that full information on the funding of the original studies (not specifically related to this analysis) can be found in the Supporting Information files. Therefore, please include the additional funding information as a Supporting Information file.

5) Please state in the Methods section whether the study had a prospective protocol or analysis plan. If a prospective analysis plan (from your funding proposal, IRB or other ethics committee submission, study protocol, or other planning document written before analyzing the data) was used in designing the study, please include the relevant document(s) with your revised manuscript as a Supporting Information file to be published alongside your study and cite it in the Methods section. A legend for this file should be included at the end of your manuscript. If no such document exists, please make sure that the Methods section transparently mentions that a study protocol is not available and describes when analyses were planned, and when/why any data-driven changes to analyses took place. Changes in the analysis, including those made in response to peer review comments, should be identified as such in the Methods section of the paper, with rationale.

---

## [Editor Report · Decision Letter 4]

25 Mar 2025

Dear Dr Ferrari, 

On behalf of my colleagues and the Academic Editor, Gilaad G. Kaplan, I am pleased to inform you that we have agreed to publish your manuscript "Alcohol intake and pancreatic cancer risk: an analysis from 30 prospective studies across Asia, Australia, Europe, and North America" (PMEDICINE-D-24-02276R4) in PLOS Medicine.

I appreciate your thorough responses to the reviewers' and editors' comments throughout the editorial process. We look forward to publishing your manuscript, and editorially there are only a few remaining points that should be addressed prior to publication. We will carefully check whether the changes have been made. If you have any questions or concerns regarding these final requests, please feel free to contact me at atosun@plos.org.

Please see below the minor points that we request you respond to:

1) Ethics statement: Please note that we cannot move the manuscript past editorial acceptance until you have provided a complete list of the institutional review boards together with the relevant approval numbers. 

2) We require you to submit the statistical analysis plan as a Supporting Information file. This document will be published alongside your study. Please be sure to cite it in the Methods section. A legend for this file should be included at the end of your manuscript.

3) It appears that you did not re-upload the STROBE checklist. Please upload it and ensure that you include the following statement in the main text: This study is reported as per the Strengthening the Reporting of Observational Studies in Epidemiology (STROBE) guideline (S1 Checklist).

4) Please note that we will try to include all funding details in the Financial Disclosure Statement. However, due to the limitations of the PDF format, we cannot guarantee that this will be possible.

Before your manuscript can be formally accepted you will need to complete some formatting changes, which you will receive in a follow up email (including the editorial points above). Please be aware that it may take several days for you to receive this email; during this time no action is required by you. Once you have received these formatting requests, please note that your manuscript will not be scheduled for publication until you have made the required changes.

PRESS

Sincerely, 

Alexandra Tosun, PhD 

Associate Editor 

PLOS Medicine